# Thylakoid protein FPB1 synergistically cooperates with PAM68 to promote CP47 biogenesis and Photosystem II assembly

Lin Zhang [1,5], Junxiang Ruan[1,5], Fudan Gao[1], Qiang Xin [1], Li-Ping Che[1], Lujuan Cai[1], Zekun Liu[2], Mengmeng Kong[3], Jean-David Rochaix [4], Hualing Mi [3] & Lianwei Peng [1] ✉

In chloroplasts, insertion of proteins with multiple transmembrane domains (TMDs) into thylakoid membranes usually occurs in a co-translational manner. Here, we have characterized a thylakoid protein designated FPB1 (Facilitator of PsbB biogenesis1) which together with a previously reported factor PAM68 (Photosynthesis Affected Mutant68) is involved in assisting the biogenesis of CP47, a subunit of the Photosystem II (PSII) core. Analysis by ribosome profiling reveals increased ribosome stalling when the last TMD segment of CP47 emerges from the ribosomal tunnel in *fpb1* and *pam68*. FPB1 interacts with PAM68 and both proteins coimmunoprecipitate with SecY/E and Alb3 as well as with some ribosomal components. Thus, our data indicate that, in coordination with the SecY/E translocon and the Alb3 integrase, FPB1 synergistically cooperates with PAM68 to facilitate the co-translational integration of the last two CP47 TMDs and the large loop between them into thylakoids and the PSII core complex.

Photosystem II (PSII) complexes in photosynthetic organisms catalyse electron transfer from water to plastoquinone to trigger photosynthetic electron transport during photosynthesis[1,2]. PSII from various species generally consists of the reaction centre (RC) composed of D1, D2, the α and β subunits of cytochrome (Cyt) $b_{559}$ and PsbI, the proximal antennae CP43 and CP47, the oxygen-evolving (OEC) and light-harvesting complexes[3]. It has been demonstrated that PSII assembly is a very complicated process that occurs in a step-wise manner[4,5]. The first step of PSII assembly is the formation of the D2-Cyt $b_{559}$ subcomplex which acts as a receptor complex for D1 and PsbI[6]. The precursor of D1 (pD1) is co-translationally integrated into the D2-Cyt $b_{559}$ subcomplex with a strict hierarchical and concerted order to first form the PSII RC. Then, the C-terminal 9 residues of pD1 are removed by CtpA (C-terminal peptidase) to form the mature D1 protein[7]. Subsequently, CP47, CP43 and other subcomplexes are sequentially assembled forming the intact functional PSII. During PSII biogenesis, at least two dozen of auxiliary proteins are required[5,8–10], such as HCF136 (High Chlorophyll Fluorescence136), LPA1 (Low PSII Accumulation 1), PAM68 (Photosynthesis Affected Mutant 68), HCF244, OHP1 (ONE-HELIX PROTEIN1), OHP2, HCF173, PsbN, CtpA, and RBD1 (Rubredoxin1). Among them, both LPA1 and PAM68 are integral thylakoid membrane proteins[11,12]. LPA1 specifically interacts with D1 and has been proposed to facilitate D1 synthesis during PSII assembly or repair[11]. In contrast, PAM68 interacts with LPA1 and several PSII core subunits and it has been proposed to be required for pD1 processing as well as PSII RC assembly in Arabidopsis[12].

Proximal antenna CP47 is an intrinsic subunit associated with 16 chlorophyll and several β-carotene molecules[13]. It contains 6 transmembrane domains (TMDs) which form three extrinsic loop regions on the lumen side of CP47. Among them, the largest loop (E loop)

[1]Development Center of Plant Germplasm Resources, College of Life Sciences, Shanghai Normal University, Shanghai 200234, China. [2]Shanghai Key Laboratory of Plant Molecular Sciences, College of Life Sciences, Shanghai Normal University, Shanghai 200234, China. [3]National Key Laboratory of Plant Molecular Genetics, CAS Center for Excellence in Molecular Plant Sciences / Institute of Plant Physiology and Ecology, Shanghai Institutes for Biological Sciences, Chinese Academy of Science, Shanghai 200032, China. [4]Departments of Molecular Biology and Plant Biology, University of Geneva, 1211 Geneva, Switzerland. [5]These authors contributed equally: Lin Zhang, Junxiang Ruan. ✉e-mail: penglianwei@shnu.edu.cn

between the fifth and sixth TMD contains about 200 amino acids (Supplementary Fig. 1a). In spinach PSII, the E loop of CP47 interacts with the luminal extrinsic proteins PsbP, PsbO, and PsbTn as well as with the luminal loop of D2. Two small membrane-intrinsic subunits PsbH and PsbL associate with CP47, and they further interact with D2 (Supplementary Fig. 1b)[14]. During the assembly process of PSII, pre-CP47 subcomplex containing CP47 and several small membrane intrinsic subunits associate with PSII RC to form the CP43-less PSII complex. Synthesis and assembly of CP47 also require the assistance of several proteins. In the absence of PsbH, synthesis of CP47 is blocked in Arabidopsis[15]. The Psb28 protein binds to the C-terminus of CP47 and the D-E loop of D1[16]. It may assist CP47 to join PSII RC. In *Synechocystis* PCC 6803, the cyanobacterial ortholog of PAM68 has been proposed to be required for the insertion of chlorophyll into CP47[17]. However, additional factors specifically required for the synthesis and assembly of CP47 most likely exist but are still unknown in higher plants.

Like CP47, a total of 37 chloroplast-encoded proteins are intrinsic thylakoid proteins and 19 of them have been demonstrated to be co-translationally inserted into the membrane[18]. PSII reaction centre protein D1, which contains five TMDs, has been well studied[19]. Before the first TMD of D1 emerges from the ribosome tunnel during translation, ribosomes bind to cpSRP54 through interaction with the ribosomal protein L4[20]. Then the ribosome nascent chain (RNC) of D1 is directed to the SecY/E translocon in the thylakoid membrane by the SRP receptor cpFtsY[20–22]. During translation, the nascent D1 chain exits from the SecY/E channel laterally and interacts with D2. It has been shown that, while the first two TMDs of D1 associate only weakly with D2, a tight interaction with D2 occurs when the fourth TMD is synthesised[23]. Besides, Alb3 and VIPP1 also interact with the SecY/E translocon during the co-translational insertion of D1 into the receptor precomplex of D2-Cyt $b_{559}$[24]. Although the PSII core subunits D2, CP43, and CP47 have been proposed to be assembled co-translationally, it is not known whether the SecY/E-cpSRP54 targeting machinery is involved in their translation and assembly and how the integration of the TMDs of these subunits into the membrane is regulated by molecular chaperones[18].

In this work, we characterized a thylakoid protein FPB1 (Facilitator of PsbB biogenesis1) that is specifically required for accumulation of PSII. We show that the rate of CP47 synthesis is reduced and formation of the pre-CP47 complex is impaired in *fpb1* and in the previously reported PSII mutant *pam68*. In *fpb1* and *pam68*, ribosome elongation pauses when the last TMD segment of CP47 emerges from the ribosomal tunnel during *psbB* translation. FPB1 interacts with CP47, PAM68, Alb3 integrase as well as with some components of SecY/E. Therefore, we propose that FPB1 and PAM68 are jointly involved in assisting the integration of one specific region of CP47 into the thylakoid membrane and the PSII core complex during the de novo assembly and/or repair of PSII.

## Results

### Accumulation of PSII is significantly reduced in the *fpb1* mutant
To identify the regulatory factors involved in the biogenesis of thylakoid protein complexes, we collected more than 500 T-DNA Arabidopsis insertion mutants from NASC (European Arabidopsis Stock Centre), in which the mutated genes encode chloroplast-localized proteins. Screening of these mutants using chlorophyll fluorescence imaging revealed that the *fpb1* mutant exhibits a low Fv/Fm ratio (Fig. 1a), which represents a proxy for the overall PSII activity. Leaf colour of the *fpb1* mutant is pale-green and the size of 4-week-old *fpb1* seedlings is much smaller compared to wild-type (WT) plants (Fig. 1a). The *fpb1* mutant harbours a T-DNA insertion in the first exon of the *FPB1* gene (Supplementary Fig. 2a). Introduction of the *FPB1* gene into *fpb1* fully rescued its phenotype with regard to plant size, leaf colour as well as the Fv/Fm ratio (Fig. 1a). Immunoblot analysis using an antibody

against FPB1 detected a protein with a molecular mass less than 15 kDa in WT and *fpb1*-complemented plants but absent in the *fpb1* mutant (Supplementary Fig. 2b). These results demonstrate that *fpb1* is a null mutant for FPB1.

Chlorophyll *a* fluorescence transients analyses showed that the minimum chlorophyll *a* fluorescence ($F_o$) was significantly higher in the *fpb1* mutant compared to WT and *fpb1*-complemented leaves, resulting in a low Fv/Fm ratio (about 0.4) (Fig. 1a and Supplementary Fig. 3a). Moreover, chlorophyll *a* fluorescence in *fpb1* decreased below the $F_o$ level 30 s after turning on the AL light and then gradually increased to $F_o$ (Supplementary Fig. 3a). This kind of chlorophyll *a* fluorescence kinetics is typical of mutants accumulating low amounts of PSII compared with PSI, such as *hcf136*, *lpa1*, and *pam68*[11,12,25]. This fluorescence pattern could be attributed to higher PSI activity relative to PSII upon initiation of photosynthetic electron flow. Indeed, absorbance kinetics of PSI reaction centre P700 at 830 nm showed that oxidation of P700+ can be induced with far-red light and a saturating pulse (SP) in far-red light background can induce the re-reduction of P700+ in *fpb1* as in WT plants (Supplementary Fig. 3b), indicating that the PSI complex is functional in the *fpb1* mutant. However, application of actinic light results in oxidation of one-quarter of total P700 (ΔA/ΔAmax) in WT and *fpb1*-complemented plants. In contrast, this light intensity leads to a complete P700 oxidation in the *fpb1* mutant (Supplementary Fig. 3b), consistent with the limited electron flow from PSII. Thus, the *fpb1* mutant is likely to be primarily affected in PSII.

Blue native (BN)-PAGE analyses of thylakoid complexes showed that PSII supercomplexes are nearly undetectable and the levels of PSII dimer, monomer, and CP43-less PSII are reduced to some extent in *fpb1* compared with WT and *fpb1*-complemented plants (Fig. 1b). Immunoblot analyses of representative subunits of photosynthetic complexes showed that accumulation of the PSII core subunits is reduced to less than or about 1/4 in the *fpb1* mutant compared with WT (Fig. 1c). The level of the PSII OEC subunit PsbO is reduced about 2-4-fold compared with WT. While the levels of PSI (PsaA and PsaD) in *fpb1* are reduced less than half relative to WT, the amounts of Cyt $b_6f$ complex (Cyt $b_6$), LHCII complex (Lhcb1), and ATP synthase (CF$_1\gamma$ and CF$_1\varepsilon$) are comparable with those of WT (Fig. 1b). These results indicate that accumulation of PSII, especially of the PSII supercomplexes, is affected in the mutant.

To investigate the biogenesis of PSII in the *fpb1* mutant, thylakoid protein complexes were separated by 2D BN/SDS-PAGE and then immunoblotted. In the WT and *fpb1*-complemented plants, PSII subunits were well detected in various PSII complexes. However, the majority of the retained PSII subunits in *fpb1* were found in the PSII monomer and CP43-less PSII complex (Fig. 1d). In particular, a trace amount of CP47 was detected at the pre-CP47 complex position in WT and *fpb1*-complemented plants but it was almost undetectable in *fpb1* (Fig. 1d). In contrast, a relative higher level of PSII RC was detected in *fpb1* compared with WT and complemented plants (Fig. 1d). This implies that formation of the pre-CP47 complex is compromised in the absence of FPB1, which in turn slows down the assembly of PSII RC into the PSII monomer, dimer and supercomplexes.

### Levels of chloroplast-encoded *psb* transcripts are not reduced but polysome association with *psbB* is enhanced in *fpb1*
To investigate whether the reduced accumulation of PSII is caused by a defect in the transcription of chloroplast-encoded *psb* genes, we analysed the accumulation of their transcripts by RNA blot hybridization. The results show that both the level and the expression pattern of *psbA*, *psbB*, *psbC*, *psbD*, *psbEFLJ*, and *psbKI* transcripts are almost identical in *fpb1* and WT plants (Supplementary Fig. 4a), suggesting that FPB1 is most probably not involved in the transcription and subsequent maturation of these PSII transcripts.

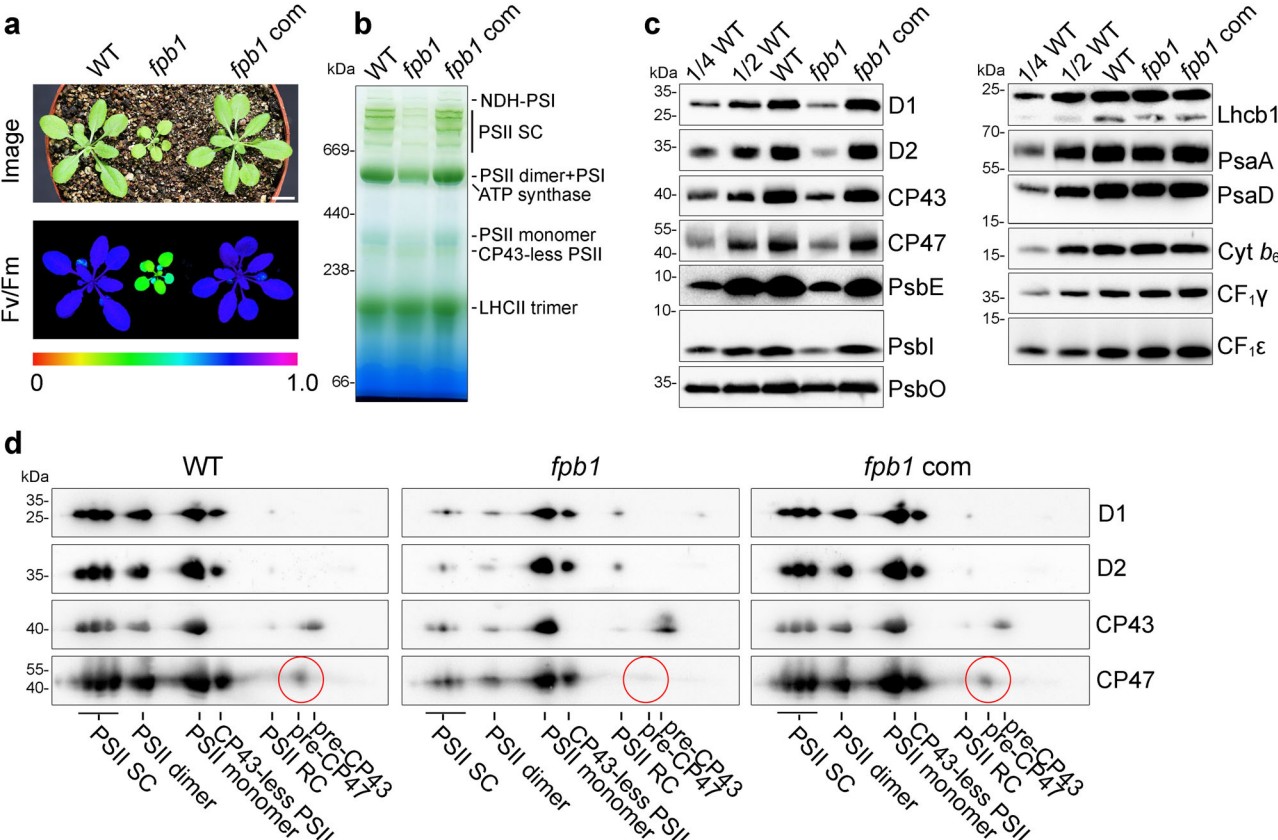

**Fig. 1 | Reduced level of PSII in *fpb1*. a** The image was taken from four-week-old plants grown in the greenhouse under about 50 µmol photons m$^{-2}$ s$^{-1}$ (top panel). Images of maximum quantum yield of PSII (Fv/Fm) were captured using a PAM imaging system. A false colour scale for Fv/Fm is shown at the bottom of the panel. *fpb1* com, *fpb1* mutant complemented with the *FPB1* genomic sequence. Bar = 1 cm. **b** Blue native (BN)-PAGE analyses of the thylakoid protein complexes from four-week-old wild type (WT), *fpb1*, and *fpb1* complemented (*fpb1* com) plants. **c** Immunodetection of representative subunits of thylakoid protein complexes.

Equal amounts of thylakoid proteins were separated by Tricine-SDS-PAGE (for PsbE, PsbI, and Cyt $b_6$) or SDS-urea-PAGE (for others) and then probed with antibodies as indicated. A series of dilutions from WT was used for rough estimation of proteins. **d** 2D BN/SDS-urea-PAGE analyses of various PSII complexes from the *fpb1* mutant. The pre-CP47 complex detected by CP47 antibody is surrounded by red circles. PSII SC, PSII supercomplexes. PSII RC, PSII reaction centre. Data are representative of two independent biological replicates. Source data are provided as a Source Data file.

Polysome association with *psb* transcripts were further examined by sucrose density gradient ultracentrifugation. For the *psbA*, *psbC*, *psbD*, *psbEFLJ*, and *psbKI* transcripts, no obvious difference in distribution of ribosomes within the gradients was found between *fpb1* and WT plants (Fig. 2a and Supplementary Fig. 4b). However, there was a significant shift of *psbB*-containing transcripts to higher molecular weight fractions in *fpb1* compared with WT (Fig. 2a). To test whether *psbB* mRNA in the higher molecular weight fractions is indeed associated with polysomes, the samples were treated with puromycin, which is a ribosome-specific disassembler. Puromycin treatment induced clear shifts of *psbB* and *psbEFLJ* transcripts towards lower molecular weight fractions in both the *fpb1* mutant and WT (Fig. 2a), indicating that the *psbB* transcripts migrating deeper into the sucrose density gradient in *fpb1* represent polysomes associated with mRNA.

## Synthesis of CP47 is reduced and PSII assembly is impaired in *fpb1*

Since polysome association with *psbB* transcripts is enhanced in the *fpb1* mutants, it is possible that the synthesis of CP47 is increased. Alternatively, elongation or termination of translation could be affected for this protein. To examine these possibilities, in vivo pulse labelling of chloroplast proteins was performed (Fig. 2b, c). Surprisingly, the rate of synthesis of CP47 was reduced to ~50% in *fpb1* compared to WT (Fig. 2b), which is not compatible with our hypothesis that translation initiation of CP47 is enhanced based on the polysome

loading results (Fig. 2a). Rather, it suggests that elongation or termination of translation is affected.

Newly synthesized thylakoid proteins were chased with cold Met for 15 and 60 min to investigate the turnover of the newly synthesized proteins. The results showed that the level of newly synthesised CP47 is not reduced after chasing for 60 min (Fig. 2d), suggesting that CP47 is stable in the thylakoid membrane once it is synthesized and the reduced rate of synthesis of CP47 is not due to enhanced turnover. In the *hcf107* mutant, synthesis of PsbH is blocked, which in turn decreased the synthesis of CP47[15]. This led us to investigate the rate of synthesis of PsbH in *fpb1*. After labelling for 20 min, thylakoid proteins were separated by Tricine-SDS-PAGE. This experiment revealed that synthesis of PsbH in *fpb1* is comparable to WT based on quantification of signal intensities (Fig. 2c), indicating that synthesis of PsbH is unaffected in *fpb1*.

Incorporation of [$^{35}$S]-Met into the D2/pD1/D1 proteins of *fpb1* was also reduced to about half relative to WT and D2/pD1 was the most heavily labelled band in *fpb1*, in contrast with WT plants in which mature D1 protein was the most highly labelled PSII subunit (Fig. 2b). Accumulation of D1 progressively increased during the chase period and about half of the pD1 protein was processed into D1 protein after a 15 min chase in *fpb1* (Fig. 2d). After a 60 min chase, mature D1 protein became the dominant form and the level of the pD1/D2 was comparable with that in WT (Lane 60 from WT and *fpb1*; Fig. 2d). These results indicate that not only the rate of synthesis of D1 is reduced but also the

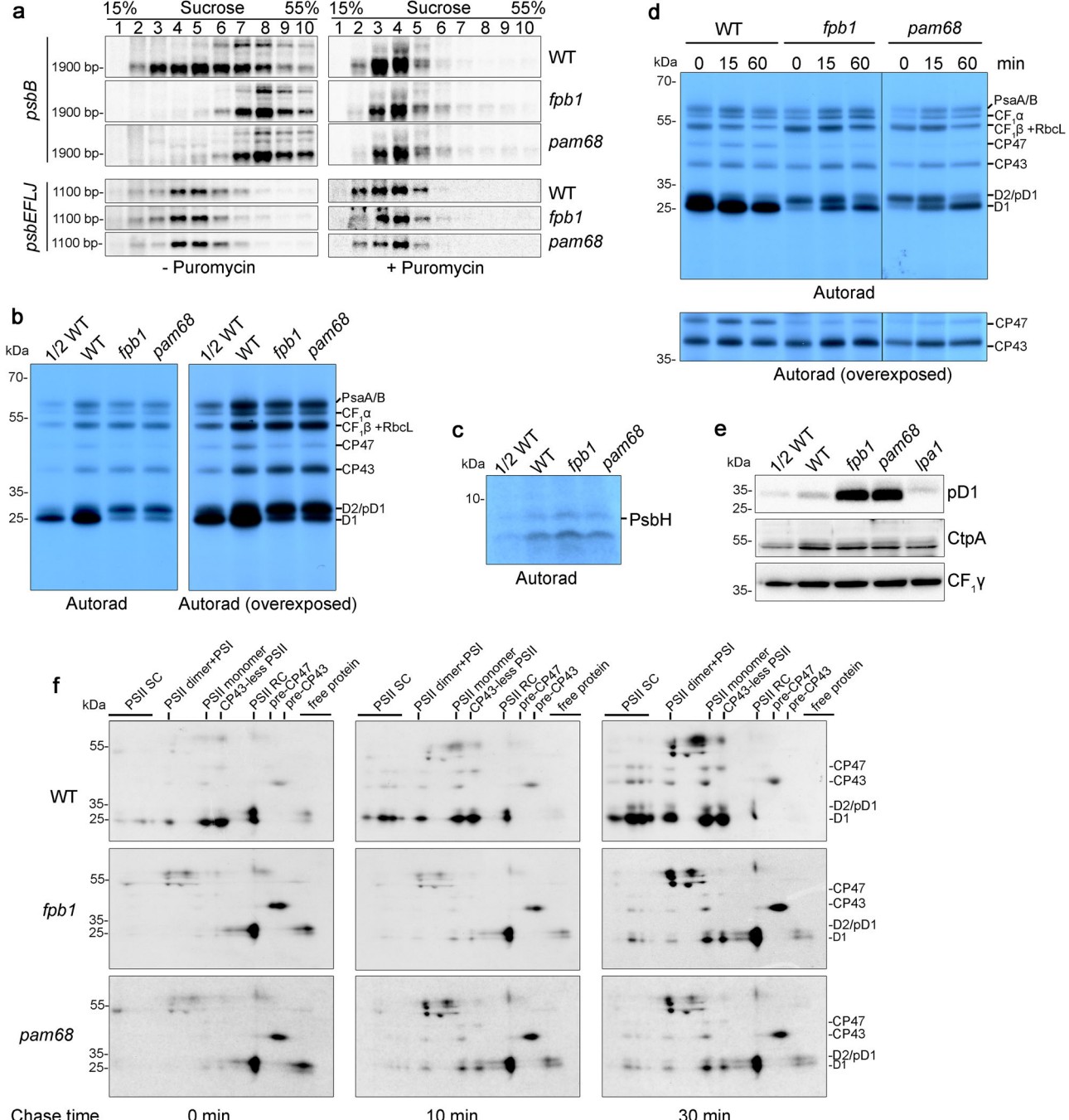

**Fig. 2 | Synthesis of CP47 is reduced and PSII assembly is retarded in *fpb1* and *pam68*. a** Polysome association studies of WT, *fpb1*, and *pam68* plants in the absence (left) and presence (right) of puromycin. Whole leaf extracts from three-week-old plants were separated by centrifugation in 15–55% sucrose density gradients. RNA isolated from the gradients was probed with DIG-labelled probes corresponding to the *psbB* and *psbEFLJ* mRNAs. Approximate sizes of main bands are shown and the corresponding band sizes can found in Supplementary Fig. 4a. **b**, **c** Protein labelling with [35S]-Met. Primary leaves of 12-day-old young seedlings were incubated with cycloheximide and then radiolabelled with [35S]-Met for 20 min. Labelled thylakoid proteins were separated using 12.5% SDS-urea-PAGE (**b**) and 16% Tricine-SDS-PAGE (**c**). The signals were detected by exposure to X-film. Overexposure from the same gel was used for higher visibility of CP47 (right in **b**). The PsbH protein was detected as a band smaller than 10-kDa as in ref. 15, in which the synthesis of PsbH (~8 kDa) was reported to be completely blocked in the *hcf107*

mutant. **d** Pulse-chase labelling of thylakoid proteins. Proteins were labelled as in (**b**) for 20 min (indicated as 0) and then chased with unlabelled Met for 15 and 60 min. Thylakoid proteins were separated and visualized as in (**b**). Overexposure of the same gel was used for detecting CP47 (bottom). **e** Immunoblot analysis of pD1 and CtpA. During illumination, 12-day-old young seedlings were frozen in liquid $N_2$ immediately and thylakoids were isolated. Equal amounts of total thylakoid proteins were separated using SDS-urea-PAGE and then probed with antibodies. $CF_1\gamma$ was used as a loading control. **f** Assembly kinetics of PSII in *fpb1*, *pam68*, and WT plants. Thylakoid proteins were labelled for 20 min (indicated as 0) and subsequently chased for 10 and 30 min. Labelled proteins were separated by 2D BN/SDS-urea-PAGE and detected using autoradiography. PSII SC, PSII supercomplexes. PSII RC, PSII reaction centre. Data are representative of two independent biological replicates. Source data are provided as a Source Data file.

processing of pD1 is significantly retarded in *fpb1*. The increased level of pD1 in *fpb1* was also confirmed by immunoblot analyses using antibodies specific against the C-terminal 9 residues of pD1 (Fig. 2e), which are removed during the pD1 processing by the enzyme CtpA[7]. It is notable that the level of CtpA is comparable in the *fpb1* and WT plants (Fig. 2e), indicating that the slower processing of pD1 in *fpb1* cannot be explained by a limited amount of CtpA enzyme.

After labelling for 20 min, synthesis of the PSII subunit CP43 and the CF$_1$β subunit of chloroplast ATP synthase as well as of RbcL (large subunit of RuBisCO) bound to thylakoids was almost unchanged in the *fpb1* mutant (Fig. 2b). Synthesis of the PSI subunits PsaA/B and the CF$_1$α subunit of chloroplast ATP synthase in *fpb1* were moderately reduced compared with WT (Fig. 2b). A slight reduction of synthesis of the PSI subunits was also reported for other mutants defective in PSII assembly such as *hcf136*, *hcf244*, *hcf173*, *ohp1*, and *ohp2*, most likely a secondary effect of PSII deficiency[25–27].

## PSII assembly is retarded in *fpb1*

In WT, about half of the D1/pD1/D2 proteins are present in the PSII RC after labelling for 20 min. During the chase, pD1 was gradually processed into mature D1 and then, together with D2, CP43, and CP47, mature D1 protein was assembled into the large PSII complexes (Fig. 2f). In contrast, in the *fpb1* mutant, newly synthesized D1/pD1/D2 proteins were mainly detected in the PSII RC after labelling for 20 min and even after a 30 min chase. Only small amounts of CP43 and D1/D2 proteins were incorporated into various PSII complexes after the 30 min chase (Fig. 2f). While CP43 was present mainly in the pre-CP43 complex, newly synthesized CP47 was predominantly detected in the PSII monomer and CP43-less PSII after a 30 min chase and no CP47 was found in the pre-CP47 complex in *fpb1* (Fig. 2f). In contrast, we reported previously that newly CP47 protein can over-accumulate in the pre-CP47 complex in *ohp1*, *ohp2*, and *rbd1* mutants, in which formation of the PSII RC complex is abolished[27,28]. These results suggest that, in the *fpb1* mutant, reduced accumulation of the pre-CP47 complex results in decreased PSII biogenesis at the stage of assembly from PSII reaction centre to CP43-less PSII and subsequent PSII monomer.

## Rate of synthesis of CP47 is also reduced in *pam68*

Previous reports showed that newly synthesized pD1 protein over-accumulates in *pam68* but not in *lpa1*, although the steady level of PSII complex is reduced to ~20% compared to WT in both mutants[11,12]. Immunoblotting using pD1 antibody and protein labelling analyses revealed a similar trend of pD1 over-accumulation for *fpb1* and *pam68* but not for *lpa1* (Fig. 2e)[12]. This led us to investigate the rate of thylakoid protein synthesis in the *pam68* mutant. Interestingly, patterns of newly synthesised thylakoid proteins in *pam68* were very similar with those in *fpb1* (Fig. 2b, c). During the 15- and 60-min chase, retarded processing of pD1 also occurred and the newly synthesized CP47 remained stable in *pam68* as in *fpb1* (Fig. 2d). Moreover, polysome association with *psbB* mRNA also shifted towards higher molecular weight fractions in the sucrose density gradient of *pam68* as observed for *fpb1* (Fig. 2a). These results suggest that FPB1 and PAM68 may be involved in a closely related process essential for CP47 synthesis, pD1 processing, and subsequent PSII biogenesis.

## Accumulation of PSII is completely abolished in the *fpb1 pam68* double mutant

Similar defects in the synthesis of the thylakoid proteins in *fpb1* and *pam68* suggested a possible genetic or biochemical link between these two mutants. To this end, a double mutant *fpb1 pam68* was generated (Fig. 3a). Since *lpa1* accumulates a similar amount of PSII as *fpb1* (Fig. 3b)[11], we also generated an *fpb1 lpa1* double mutant. Compared with WT and single mutants, growth of *fpb1 pam68* was decreased with yellow pale leaves and the Fv/Fm ratio was close to 0 (Fig. 3a). In contrast, the *fpb1 lpa1* double mutant had a similar growth rate, leaf colour, and Fv/Fm ratio as the *fpb1* and *lpa1* single mutants (Fig. 3a). In the *fpb1 pam68* double mutant, the levels of the PSII core subunits were markedly decreased, less than 1/16 of WT or below the detection level (Fig. 3b). Similar levels of PSII subunits were detected in the *fpb1 lpa1* double mutant as in the *lpa1* single mutant, consistent with their growth and photosynthetic phenotypes (Fig. 3b).

Although PSII complex accumulation is almost completely abolished in the *fpb1 pam68* double mutant, it may accumulate some assembly intermediates of PSII. In order to increase the level of PSII subunits, the seedlings were grown under very low light irradiance to mitigate photodamage to chloroplasts. Indeed, trace amounts of D1 and D2 were detected in the PSII RC complex and CP43 was present in the pre-CP43 assembly intermediate in *fpb1 pam68* (Fig. 3c). Similar to *fpb1*, *pam68* also accumulates low level of pre-CP47 complex compared with WT plants (Fig. 3c). Interestingly, the pre-CP47 complex is completely absent in the *fpb1 pam68* double mutant and the retained CP47 protein migrates in several complexes with molecular masses similar to PSII supercomplexes and monomer (Fig. 3c). Since no D1, D2, and CP43 were detected in the PSII supercomplexes and monomer position in *fpb1 pam68*, CP47 seems to be associated with high molecular weight unknown complexes which may represent protein aggregates.

Taken together, our results imply that the rate of synthesis of CP47 is reduced and pD1 processing is slowed down in the *fpb1* and *pam68* mutants, indicating that FPB1 and PAM68 perform a role in a closely related process during PSII assembly. The similar profiles of protein labelling and polysome association with *psbB* mRNA in *pam68* and *fpb1* as well as the complete absence of pre-CP47 complex in the *fpb1 pam68* double mutant indicate that the CP47 protein could be the primary target of these two auxiliary factors.

## FPB1 is a thylakoid membrane protein with unknown function

FPB1 is a 181-amino acid protein with unknown function. Orthologs were found in terrestrial plants and the green alga *Chlamydomonas reinhardtii*, but not in cyanobacteria (Fig. 4a). Arabidopsis FPB1 was predicted to contain a transit peptide of 36 amino acids and two TMDs (Fig. 4a). Co-localization of the FPB-GFP signal and chlorophyll autofluorescence confirmed that FPB1 is exclusively localized in chloroplasts (Fig. 4b). Consistently, immunoblot analysis of chloroplast stromal and thylakoid proteins showed that FPB1 is localized in the thylakoid membrane (Fig. 4c) and enriched in the stroma lamellae (Fig. 4d), where the assembly of PSII occurs. Since FPB1 is predicted to contain two TMDs, it is likely an intrinsic membrane protein. Indeed, incubation of thylakoid membranes with salts did not release FPB1 from thylakoids. Although a trace amount of FPB1 was released from thylakoids with NaOH, the majority of FPB1 still remained in thylakoids (Fig. 4e). In contrast, the lumenal protein PsbO and peripheral protein CF$_1$β were partially released from thylakoids upon incubation with Na$_2$CO$_3$, CaCl$_2$, and NaOH (Fig. 4e). These results indicate that FPB1 is an intrinsic thylakoid membrane protein mainly located in chloroplast stroma lamellae.

To assess the topology of FPB1, thylakoid membranes were subjected to trypsin digestion (Fig. 4f). If the N- and C-termini of FPB1 are exposed to the chloroplast stroma (Topology 1 in Fig. 4g), trypsin digestion will generate one fragment containing two TMDs which could not be detected with the immunoblot assay because the antibody against FPB1 was raised against its N-terminal soluble part (Fig. 4g). In contrast, if the N- and C-termini of FPB1 are present in the thylakoid lumen (Topology 2 in Fig. 4g), trypsin digestion will generate two fragments. The larger one containing the N-terminus and the first TMD could potentially be detected as a 10-kDa fragment with the antibody used. However, immunoblot analysis of the trypsin-digested thylakoid membrane did not detect any fragment (Fig. 4f), in agreement with the prediction of Topology 1. Thus, we conclude that N- and

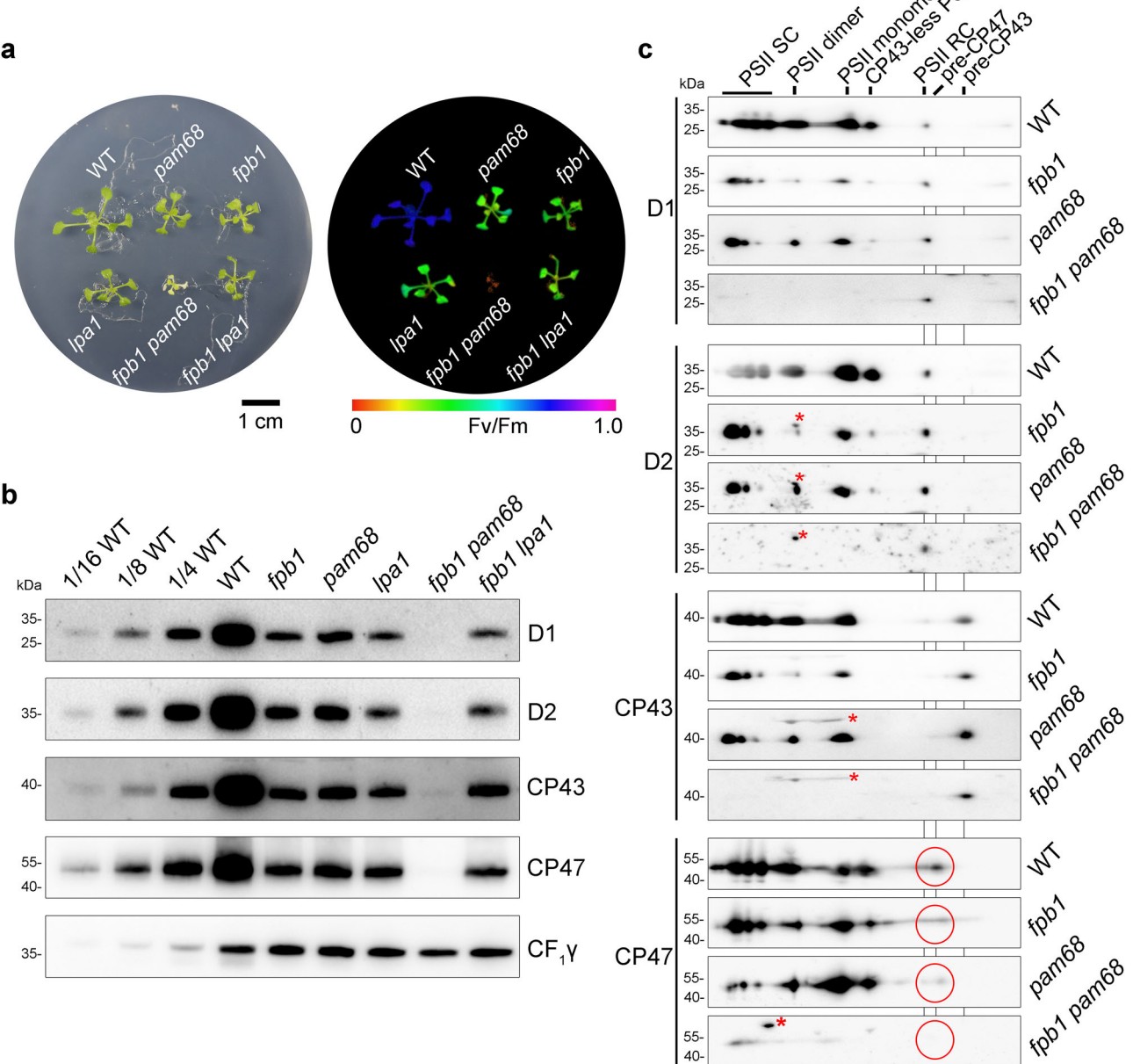

**Fig. 3 | Characterization of the *fpb1 pam68* double mutant. a** Phenotypes (left) and Fv/Fm value (right) of 3-week-old *fpb1, pam68, lpa1, fpb1 pam68, fpb1 lpa1* and WT plants grown on MS medium containing 3% sucrose under low light (20−30 μmol photons m$^{-2}$ s$^{-1}$). Fv/Fm values are indicated on a false colour scale. **b** Accumulation of PSII core subunits in three-week-old mutant and WT plants. Equal amounts of thylakoid proteins were loaded together with a series of dilutions of WT samples as indicated and separated by SDS-urea-PAGE. Immunoblotting with CF$_1$γ antibody was used as a loading control. **c** Formation of the PSII assembly complexes in three-week-old *fpb1, pam68, fpb1 pam68*, and WT plants. To alleviate photodamage of the *fpb1 pam68* double mutant, all the genotypes were grown under very low light irradiance (-5 μmol photons m$^{-2}$ s$^{-1}$). Thylakoid proteins were separated by 2D BN/SDS-urea-PAGE and probed with antibodies against D1, D2, CP43, and CP47. Non-specific signals are indicated by red asterisks. The pre-CP47 complexes detected with CP47 antibody are circled in red. PSII SC, PSII super-complexes. PSII RC, PSII reaction centre. Data are representative of two independent biological replicates. Source data are provided as a Source Data file.

C-termini of FPB1 are located on the stromal side of the thylakoid membrane.

### FPB1 interacts with PAM68, Alb3 and SecY1

Immunoblotting analyses detected elevated levels of FPB1 and PAM68 in various mutants compared with WT plants. PSII assembly factors LPA1, HCF136, and HCF244 also stably accumulated in the *fpb1* and *pam68* mutants (Supplementary Fig. 5a), suggesting that stability of these PSII assembly factors is not dependent on either of the corresponding missing proteins. To test whether FPB1 interacts with PAM68 or other proteins in vivo, thylakoid protein complexes were separated

by sucrose density gradient ultracentrifugation (Supplementary Fig. 5b). In WT, FPB1 was distributed in the fractions 5 to 11, suggesting that FPB1 can form a complex with a molecular mass similar to that of LHCII trimer. PAM68 was distributed in fractions 4 to 12. In the absence of PAM68 there was surprisingly a slight shift of FPB1 towards the bottom of the gradient whereas in the absence of FPB1 there was a similar shift (Supplementary Fig. 5b). These results imply that additional proteins are present in the PAM68- and FPB1-containing complexes.

Co-immunoprecipitation was performed to investigate the interacting proteins of FPB1 and PAM68. As negative controls, the proteins

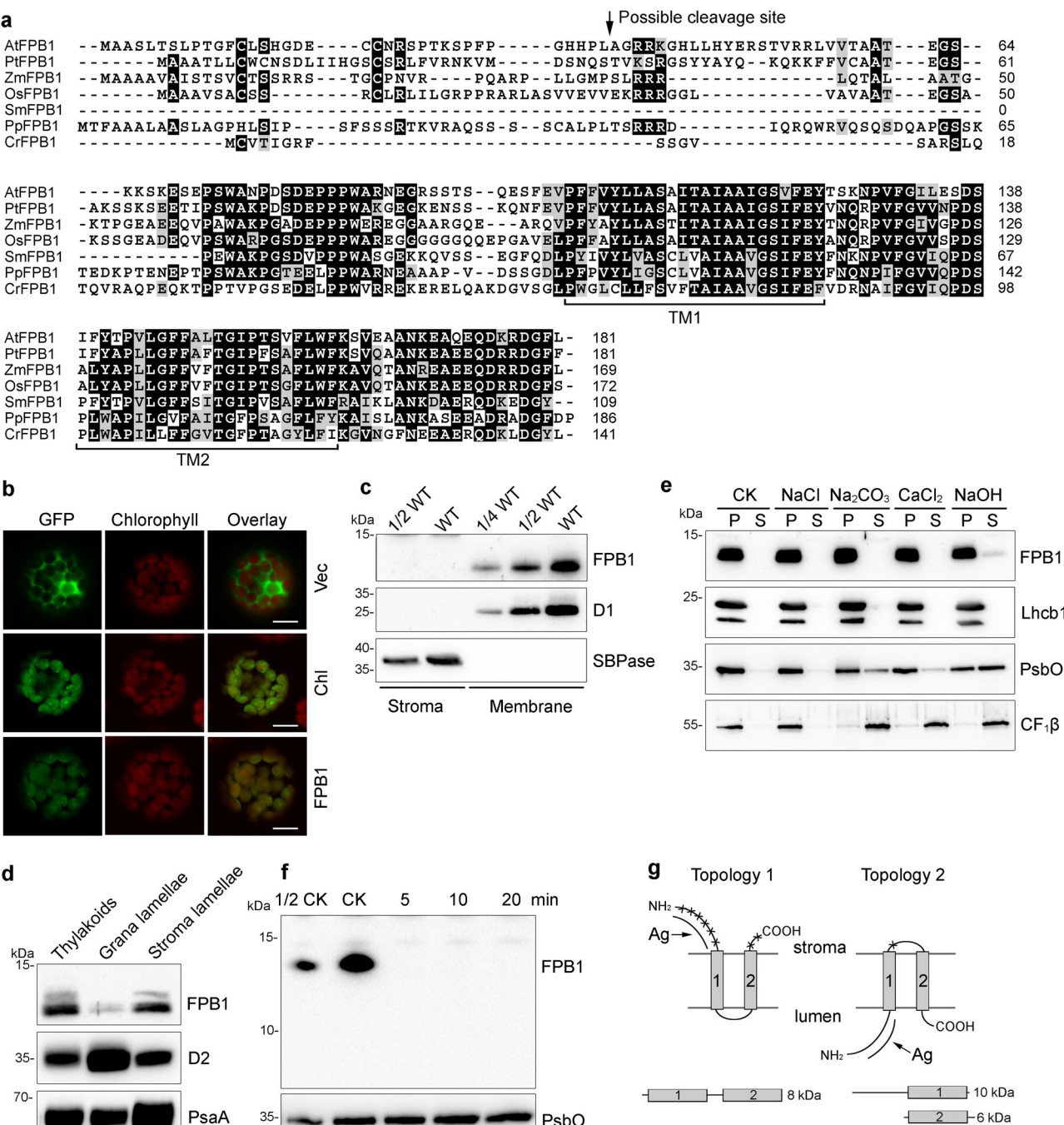

**Fig. 4 | Characterization of the FPB1 Protein. a** Sequence alignment of FPB1 and its orthologs from several photosynthetic eukaryotes. The two predicted transmembrane domains (TM1 and TM2) are indicated. The putative cleavage site for the chloroplast transit peptide of AtFPB1 is indicated by an arrow. **b** Localization of FPB1 in Arabidopsis protoplasts. Vec, empty vector control expressing only GFP protein. Chl, chloroplast localization control expressing RbcS-GFP fusion protein. FPB1, FPB1-GFP fusion protein. Bars = 10 μm. **c** Immunoblot analysis of FPB1. Intact chloroplasts isolated from WT plants were fractionated into stromal and membrane fractions, which were further analysed by immunoblotting with FPB1 antiserum. Antibodies against D1 and SBPase (fructose−1,6-bisphosphatase) were used as controls. **d** Localization of FPB1 in thylakoids. Thylakoids were fractionated into grana and stroma lamellae and immunoblot analysis was performed using antibodies against FPB1, D2, and PsaA, respectively. **e** Salt and alkali washing of thylakoid membranes. WT thylakoids were incubated without (CK) or with 0.25 M NaCl, 0.2 M $Na_2CO_3$, 1 M $CaCl_2$, and 0.1 M NaOH for 30 min on ice. Soluble and membrane fractions were separated by centrifugation and immunoblotted with FPB1, Lhcb1 (integral membrane marker), PsbO (peripheral membrane protein marker exposed to the lumenal side of thylakoids), and $CF_1\beta$ (peripheral membrane protein marker exposed to the stromal side of thylakoids) antisera. **f** Trypsin digestion analysis. Thylakoids were treated with 10 μg/mL trypsin for 0 (CK), 5, 10, and 20 min on ice. The proteins were then probed with antibodies against FPB1 and PsbO (lumenal proteins that are not accessible to the enzyme), respectively. **g** Two possible topologies of FPB1 in thylakoids. Cleavage sites in FPB1 by trypsin are indicated by asterisks. Ag represents the FPB1 region used to raise the antibody. A total of three proteolytic fragments in the two topologies are indicated but only the 10-kDa fragment with Topology 2 can be detected with FPB1 antiserum. SDS-urea-PAGE (**c**−**e**, PsbO in **f**) and Tricine-SDS-PAGE (FPB1 in **f**) were used to separate proteins. Data are representative of two independent biological replicates (**b**−**f**). Source data are provided as a Source Data file.

were also co-purified with preimmune serum (Pre) and the antibody against YCF4, a nonessential PSI assembly factor containing one TMD (Fig. 5a–c)[29]. A trace amount of FPB1 was detected in the PAM68-precipitate and PAM68 was barely detected in the FPB1-precipitate (Fig. 5a). Two bands corresponding to FPB1 were detected under denaturing conditions in this experiment, suggesting that FPB1 may undergo post-translational modifications. PSII subunit D1, D2, and CP47 but not CP43 were copurified with FPB1 and PAM68 (Fig. 5a), indicating that FPB1 and PAM68 associate with D1, D2, and CP47 in vivo. Alb3 belongs to the YidC/Oxa1/Alb3 protein family and acts as an integrase for the integration of membrane proteins into thylakoids[30]. It is also co-immunoprecipitated with both FPB1 and PAM68 (Fig. 5a). To further investigate their potential interaction in vivo, thylakoids were cross-linked and only the proteins localized closely and efficiently cross-linked with PAM68 and FPB1 can be immunoprecipitated. As shown in Fig. 5b, FPB1 was well detected in the PAM68-precipitate and vice versa. Only CP47 was co-purified with PAM68 and FPB1 (Fig. 5b), indicating that FPB1 and PAM68 directly interact or localized close to CP47. By contrast, Alb3 can be co-immunoprecipitated with FPB1 but not with PAM68 (Fig. 5b), suggesting that Alb3 and PAM68 are not directly interacting or cross-linking between them is less efficient.

Split-ubiquitin yeast two-hybrid (Y2H) assays further showed that FPB1 physically interacts with CP47 and PAM68 but not with CP43, PsaA and YCF4 (Fig. 5d). We also performed a bimolecular fluorescence complementation (BiFC) analysis in Arabidopsis protoplasts. Strong YFP signals were observed in protoplasts expressing FPB1-YFP[N] and PAM68-YFP[C] as well as in protoplasts expressing PAM68-YFP[N] and FPB1-YFP[C] (Supplementary Fig. 6a). Although PAM68-YFP[C] is over-

accumulated in protoplasts, no YFP signals were observed in protoplasts expressing YFP[N] and PAM68-YFP[C] as well as in protoplasts expressing YCF4-YFP[N] and PAM68-YFP[C] (Supplementary Fig. 6b, c). In addition, the interaction of Alb3 with FPB1 and PAM68 was detected in Y2H and BiFC assays (Fig. 5d; Supplementary Fig. 6a, d), indicating that Alb3 associates with FPB1 and PAM68 in vivo.

D1 and D2 were co-purified by immunoprecipitation with FPB1 and PAM68 antibodies using not cross-linked thylakoids, suggesting that FPB1 and PAM68 are associated with a putative complex. To identify its components, the final precipitates were subjected to mass spectrometric (MS) analyses. PSII core subunits including CP43 and OEC subunits were detected in the precipitates obtained with pre-serum, and with FPB1, PAM68 and YCF4 antibodies (Supplementary Data 1), implying that these samples were contaminated by trace amount of PSII. However, two low molecular weight PSII subunits PsbH and PsbL associated with CP47 were only detected in the FPB1-precipitate and PsbL was also detected in the PAM68-precipitate (Supplementary Fig. 1b; Data 1). These results suggest that FPB1 and PAM68 associate with putative PSII assembly intermediates, which may contain D1, D2, CP47 as well as PSII subunits that are located near the CP47 subunit (Fig. 5a). In addition, a dozen of PSII assembly or repair factors were detected or enriched in the FPB1- and/or PAM68-precipitates, such as LPA1, LPA2, LQY1, MET1, LTO1, CYP38, and HHL1 (Supplementary Data 1), which are involved in various steps of PSII assembly or repair[8–10]. LPA1 and LPA2 are required for biogenesis of D1 and CP43, respectively[11,31]. Interaction of FPB1 and PAM68 with these two assembly factors was previously reported[12] and also shown by our Y2H and BiFC assays (Fig. 5d; Supplementary Fig. 6a). These results suggest that the different steps of PSII assembly occur close to each other

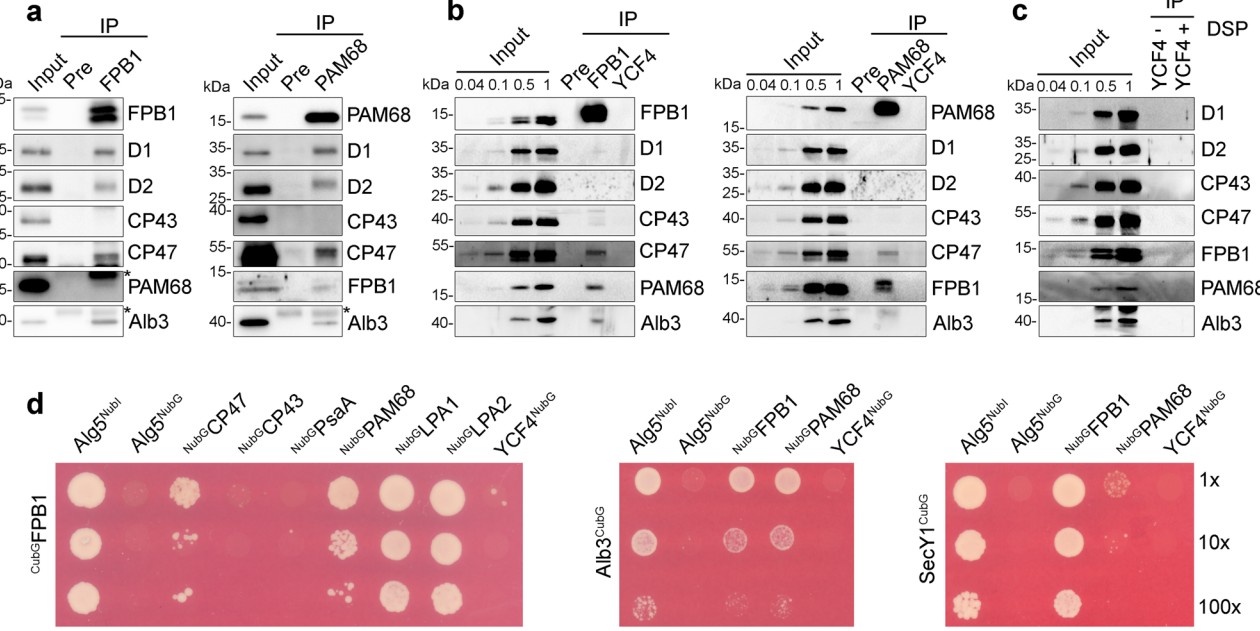

**Fig. 5 | Interaction of FPB1 and PAM68 with chloroplast proteins. a** Co-immunoprecipitation of FPB1/PAM68-containing assembly intermediates. WT thylakoids were solubilized with 1% β-DM and incubated with CNBr-activated agarose coupled with preimmune serum (Pre) as well as purified FPB1 and PAM68 antibodies. Co-immunoprecipitated proteins were separated by SDS-PAGE and then immunoblotted with various antibodies. Thylakoids corresponding to 0.1 μg chlorophyll (IB for D1, D2, CP43, and CP47) or 2.5 μg chlorophyll (IB for FPB1, PAM68, and Alb3) were used as reference of the blots. IgG eluted from the beads or unknown proteins are indicated with asterisks. **b** Immunoprecipitation of FPB1/PAM68-crosslinked proteins. Thylakoid proteins were crosslinked by DSP and then solubilized with SDS. The samples were immunoprecipitated and analysed as in **a**. **c** Immunoprecipitation of crosslinked (+ DSP) and not crosslinked (-DSP) proteins

with CNBr-activated agarose coupled with purified YCF4 antibody. Immunoblots were performed as in (**a** and **b**). The migration of PAM68 compared to the 15-kDa molecular weight marker is slightly different in SDS-PAGE (a-c) and SDS-urea-PAGE (Supplementary Figs. 5 and 6c). **d** Split-ubiquitin yeast two hybrid assays for protein-protein interactions. The Cub was fused to the N-terminus of mature FPB1 (Cub-FPB1) as well as the C-terminus of Alb3 (Alb3-Cub) and SecY1 (SecY1-Cub) as baits. Other proteins were fused to NubG at their C- or N-termini as preys. Bait and prey constructs were cotransformed into yeast NMY32 strains and the positive clones were grown on the SD-leu-trp-his-ade medium for 2–4 days at 28 °C. Alg5-NubI and Alg5-NubG were used as positive and negative controls, respectively. A series of dilutions were spotted as indicated. Data are representative of two independent biological replicates. Source data are provided as a Source Data file.

within a putative PSII assembly/repair scaffold. When PSII assembles or is repaired in this scaffold step by step, PSII assembly/repair factors may dynamically associate with or be released from the scaffold at various specific steps. Through this mechanism, PSII is efficiently assembled or repaired avoiding long distance transport of different PSII assembly intermediates along thylakoid membranes.

In chloroplasts, the SEC1 translocase works in conjunction with Alb3 to integrate membrane proteins into thylakoids[32]. Its component SecY1 and Alb3 were readily detected in the FPB1- and PAM68-precipitates but not in the Pre-serum and YCF4-precipitate by MS analysis (Supplementary Data 1). In addition, the SEC1 translocase interaction partners SecA1, FtsY, and cpSRP54 were also detected in the FPB1- and/or PAM68-precipitates (Supplementary Data 1). Y2H assays showed that SecY1 strongly interacts with FPB1 and very weakly with PAM68 (Fig. 5d). Although no interaction between SecY1 and the two assembly factors FPB1 and PAM68 was detected by BiFC, FPB1 and PAM68 interact with FtsY, SecA1, and cpSRP54 in this assay (Supplementary Fig. 6a). These results indicate that FPB1 interacts with components of SEC1 and Alb3 integrase in vivo and this interaction is likely required for the co-translational targeting and membrane integration of CP47. PAM68 may participate in this process by directly or indirectly associating with SEC1 and Alb3.

CP47 contains six transmembrane helices and its insertion into thylakoids occurs co-translationally[18]. It is possible that the ribosome machinery responsible for CP47 synthesis is also associated with the FPB1/PAM68/SEC1/Alb3 complex. Indeed, among the 58 chloroplast ribosomal proteins, 44 and 34 subunits were detected in the FPB1- and PAM68-precipitates, respectively, by MS analysis (Supplementary Data 1). Although 16 and 2 ribosomal proteins were found in the Pre-serum and YCF4-precipitates, respectively, most of them reacted only weakly. These results indicate that the ribosome machinery responsible for CP47 synthesis is also associated with FPB1 and PAM68.

**Translational elongation of *psbB* is impaired in *fpb1* and *pam68***
Polysome association with *psbB* transcript is enhanced but the CP47 synthesis rate is reduced in the absence of FPB1 and PAM68 (Fig. 2a, b), suggesting that translation of *psbB* is affected in *fpb1* and *pam68*. To further clarify the role of FPB1 and PAM68 in *psbB* mRNA translation and concomitant CP47 integration, ribosome profiling was performed. In the first experiment, two replicate samples of WT, *fpb1*, *pam68*, and *lpa1* plants were analysed. The *lpa1* mutant defective in D1 assembly was used as a control in this assay. For the third replicate, only one sample of WT, *fpb1*, and *pam68* plants was used to confirm ribosome profiling in these two mutants. In addition, RNA-Seq analyses of the samples used for Ribo-Seq were performed showing that the abundance of the chloroplast mRNAs of the PSII subunits in the *fpb1*, *pam68*, *lpa1* mutants is slightly higher or lower compared with WT (Fig. 6a; Supplementary Data 2), consistent with the RNA gel-blot hybridization analyses (Supplementary Fig. 4a)[11,12].

We obtained 50 to 84 million footprint reads in our ribosome sequencing libraries for three independent biological replicate samples (Supplementary Data 2). The ribosome footprints showed a peak at 31–33 nucleotides (Supplementary Fig. 7a). The three-nucleotide periodicity corresponds to the translocation of ribosomes along the mRNA three nucleotides at a time[33]. Our data show the 3-nucleotide periodicity expected for ribosome prints and the footprints coverage is largely restricted to the open reading frames (Supplementary Fig. 7b, c). Ribo-Seq analyses showed that the most obvious difference in the abundance of ribosome footprints between *fpb1*, *pam68*, and WT plants occurs for *psbB*. Its ribosome footprints in *fpb1* and *pam68* are about twice higher than in WT plants (Fig. 6b), which is consistent with the enhanced polysome association of *psbB* transcript in these two mutants (Fig. 2a). As expected, the *psbB* ribosome footprint abundance is comparable between *lpa1* and WT plants (Fig. 6b).

However, a significant decrease of *psbA* ribosome footprints was detected in *lpa1* (Fig. 6b), which is consistent with the reduced synthesis rate of D1 in this mutant[11,12]. No obvious difference in the abundance of *psbA* footprints was observed in *fpb1*, *pam68*, and WT plants (Fig. 6b). This observation is consistent with our conclusion that D1 biogenesis is not the primary target of PAM68 and FPB1. Together, these ribosome profiling data are supported by our biochemical and genetic analyses and thus can be used for further analysis.

To further investigate the enhanced *psbB* ribosome occupancy in *fpb1* and *pam68*, the distribution of ribosome footprints along the *psbB* mRNA was analysed. Five major peaks were detected in the distribution (Fig. 6c–e), suggesting the positions of paused ribosomes along the mRNA. Since the translation initiation of thylakoid protein was proposed to occur in the chloroplast stroma[18], the first peak at the extreme 5′ end of *psbB* mRNA should correspond to the CP47 translation initiation site. The positions of the second to fifth peaks correspond to positions near the 2nd, 4th, 5th, and 6th TMD of CP47, suggesting that ribosomes pause after each of these TMD segments emerges from the ribosomal tunnel during translation. Our results show that, throughout the entire mRNA, the read coverage in *fpb1* and *pam68* is higher than in WT (upper lanes in Fig. 6c–e), which is consistent with the higher ribosome occupancy on *psbB* mRNA detected in *fpb1* and *pam68* (Fig. 6b). However, after normalization of ribosome footprint abundance, the fifth peak in *fpb1* and *pam68* is reproducibly higher than in WT in all three replicate samples (bottom lanes in Fig. 6c–e). In contrast, the read coverage in *lpa1* is similar to that in WT and the height of the fifth peak in *lpa1* is comparable to WT (Supplementary Fig. 8). Other peaks, especially the third and fourth peaks, in the *fpb1*, *pam68*, and *lpa1* mutants are slightly increased, reduced or similar compared with those in WT (Fig. 6c–e; Supplementary Fig. 8), which is probably due to the quality of our Ribo-Seq.

To analyse the change in ribosome pausing between the different mutants and WT throughout the whole chloroplast genome, we performed a plastid genome-wide analysis (Supplementary Data 3). The results showed that, for the relative height of the *psbB* fifth peak (*psbB*-V), the ratios of *fpb1* and *pam68* to WT are $1.310 \pm 0.086$ and $1.420 \pm 0.041$, respectively. Only two peaks (*psbA*-V and *psbZ*-II) had higher relative heights in *fpb1* or *pam68* than the fifth peak of *psbB* (Supplementary Fig. 9). For the fifth peak of *psbA* (*psbA*-V), a higher relative height compared with WT was observed in *fpb1*, *pam68*, *lpa1* as well as our previous reported *rbd1* mutant[28]. This peak corresponds to the region of *psbA* mRNA encoding the third TMD, suggesting ribosome pausing when the third TMD of D1 emerges from the ribosomal tunnel. Interestingly, the rate of synthesis of D1 is reduced in all these four mutants (Fig. 2b)[11,28]. It is possible that *psbA* translation at this position is secondarily feed-back regulated in the absence of FPB1 and PAM68. Alternatively, FPB1 and PAM68 are directly involved in D1 synthesis, a possibility we cannot exclude. For the second peak of *psbZ* (*psbZ*-II), only *fpb1* had relative higher heights compared with WT (Supplementary Fig. 9). The genome-wide analysis also detected several peaks with relative slightly higher or lower heights in *fpb1* and *pam68* compared with WT (Supplementary Fig. 9). This may be due to the limitations of the Ribo-Seq technology but we cannot exclude the possibility that translation of these genes is directly or indirectly affected in the absence of FPB1 and PAM68. However, our genetic and biochemical results indicate that *psbB* is the main target of FPB1 and PAM68 and the reproducibly increased relative height of the fifth peak indicates increased ribosome stalling when the 6th transmembrane segment of CP47 emerges from the ribosomal tunnel in *fpb1* and *pam68*. The region between the 5th and 6th transmembrane segments of mature CP47 contains the E loop with 192 residues in the thylakoid lumen (Supplementary Fig. 1). Thus, the 5th and 6th transmembrane segments and the E loop need to cross thylakoid membrane after they

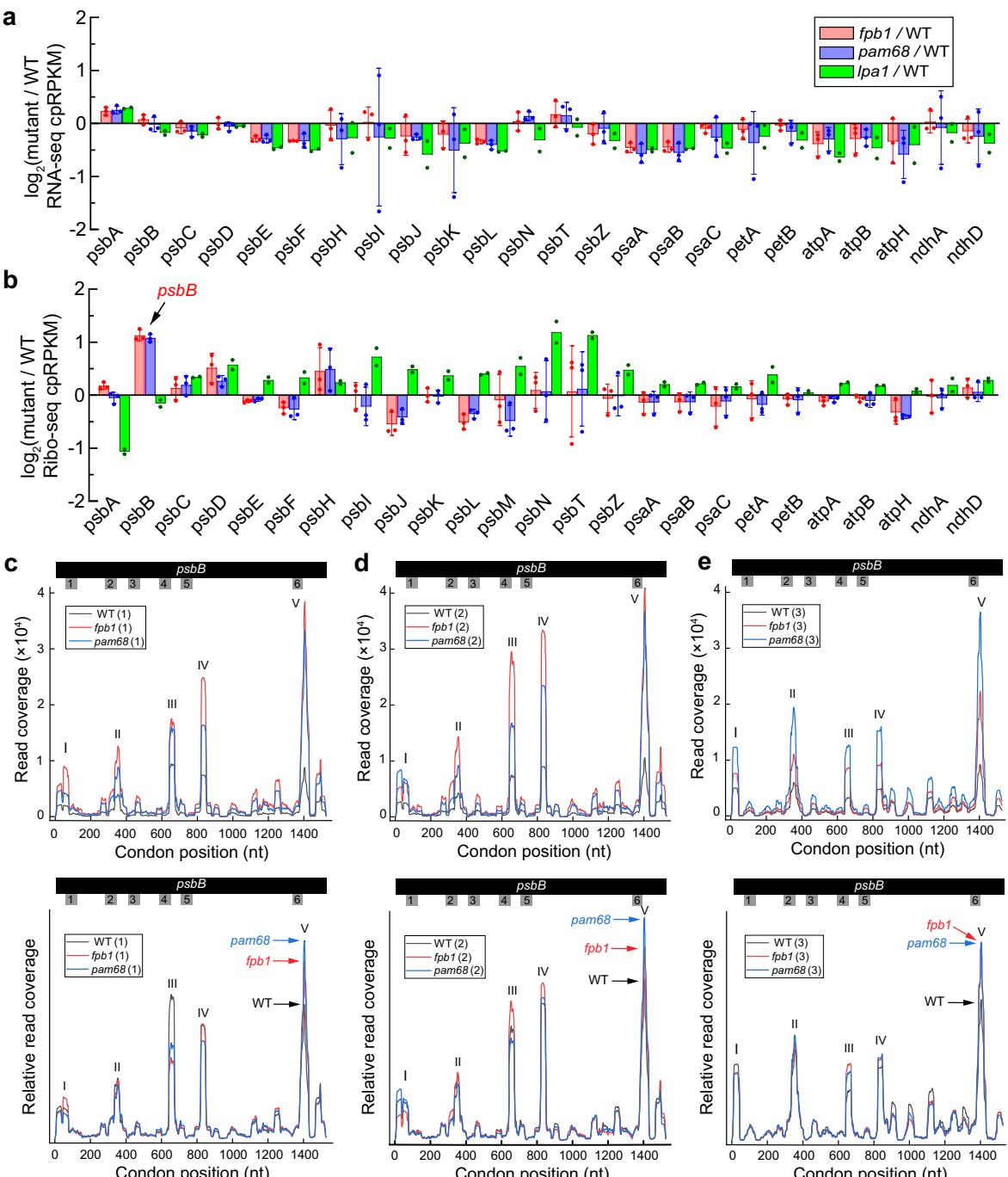

**Fig. 6 | RNA-seq and Ribo-seq analyses of chloroplast gene expression.** RNA-seq analysis of chloroplast transcript abundance (**a**) and Ribo-seq analysis of ribosome footprint abundance (**b**) in the *fpb1*, *pam68*, and *lpa1* mutants. Ratios of RNA-seq and Ribo-seq reads in the mutants relative to the wild type (WT) plants for all of the chloroplast PSII genes (For RNA-seq, the *psbM* gene was excluded because its number of reads is the lowest among PSII genes and is less than 200 in the samples of the first experiment) and of several representative chloroplast genes encoding subunits of other thylakoid protein complexes are shown. The values are the mean with standard deviations from three biological replicates of WT, *fpb1*, and *pam68* (Supplementary Data 2) and the mean with single dots from two replicates of *lpa1*. The data are expressed as cpRPKM (reads per kilobase in the ORF per million reads mapped to total chloroplast ORFs) and plotted in log₂ scale on the Y-axis. **c**–**e** Distribution of ribosome footprints along the *psbB* ORF in three replicates of WT, *fpb1*, and *pam68* mutant plants. The Y-axis shows the total reads at each position from the three genotypes (upper) or normalized based on the total reads (bottom). TMD-encoding positions on the *psbB* mRNA are indicated by Arabic numerals. Source data are provided as a Source Data file.

emerge from the ribosomal tunnel and FPB1/PAM68 may play critical roles in this process.

We also compared our results with two Ribo-seq data sets from Arabidopsis and three Ribo-seq data sets from maize published by Chotewutmontri and Barkan[34]. We found that the *psbB* peaks I, II and V were well detected (Supplementary Fig. 10), indicating that *psbB* pausing at these sites is conserved in Arabidopsis and maize. In addition, the alignment of CP47 and *psbB* mRNA sequences showed that not only the CP47 protein but also the *psbB* sequences from Arabidopsis and maize are well conserved (Supplementary Fig. 11). Thus,

conserved *psbB* mRNA features in different species maybe also contribute to ribosome pausing at the same specific sites during translation.

## Discussion

In this study, we identified a nucleus-encoded chloroplast protein termed FPB1 that is involved in PSII assembly in higher plants. While this work was under review, another group reported that the same protein as FPB1, named DEAP2, is involved in CP47 biogenesis[35]. Both our study and that of Keller et al. show that FPB1/DEAP2 acts in concert with PAM68 in the assembly of the CP47 with the PSII reaction centre to form the CP43-less PSII. Both studies also agree that the *fpb1/deap2* and *pam68* mutants display similar defects in PSII accumulation and assembly and that the absence of both proteins leads to a loss of intact PSII and the inability of double mutant to grow photoautotrophically (Fig. 3)[35]. We further show that the rate of synthesis of CP47 is reduced in *fpb1* and *pam68* and the functional pre-CP47 complex is undetectable in the *fpb1 pam68* double mutant, indicating that FPB1 synergistically cooperates with PAM68 to assist CP47 biogenesis in chloroplasts (Figs. 2b and 3). FPB1 interacts with PAM68 and they associate with the ribosome machinery, the SecY/E translocon, and Alb3 integrase (Fig. 5; Supplementary Fig. 6). Together with the increased ribosome stalling detected when the last TMD segment of CP47 emerges from the ribosomal tunnel in *fpb1* and *pam68* (Fig. 6c–e), these results indicate that FPB1 and PAM68 most likely facilitate the co-translational integration of the last two TMDs and the connecting loop of CP47 into thylakoids. Because genome-wide analysis detected several changes in ribosome pausing in *fpb1* or *pam68*, we cannot completely exclude the possibility that FPB1 and PAM68 play a minor role in the expression of other chloroplast genes (Supplementary Fig. 9).

The processing of pD1 is significantly retarded in *fpb1* and *pam68*, resulting in an over-accumulation of pD1 in thylakoids (Fig. 2d, e). However, this appears to result indirectly from the deficiency of CP47 during PSII assembly. The PsbH subunit is a conserved phosphoprotein with a single TMD that directly interacts with CP47 in thylakoids (Supplementary Fig. 1b)[14]. Synthesis of CP47 is severely impaired in mutants defective in *psbH* expression in *Synechocystis* PCC 6803 and Arabidopsis[15,36]. In these mutants, an elevated level of the pD1 protein was detected, as observed for the *psbB* deletion mutants in *Synechocystis*[15,36,37]. In addition, PsbH was also suggested to be involved in CES (for control by epistasy of synthesis) regulation of *psbB* in Chlamydomonas[38]. These observations suggest that PsbH plays conserved roles in CP47 synthesis and show that efficient pD1 processing correlates with the synthesis and accumulation of CP47/PsbH[37]. In our study, after radiolabeling, almost equal amounts of mature D1 and pD1 were found in the WT PSII RC subcomplex but only mature D1 protein was detected in the CP43-less PSII and other intact PSII complexes (Fig. 2f), supporting the conclusion that the binding of CP47 to PSII RC accelerates the processing of pD1 observed in *Synechocystis*[37]. However, CP47 and PsbH do not directly interact with D1 and it remains enigmatic how CP47 facilitates maturation of pD1 (Supplementary Fig. 1b)[14]. It is likely that the binding of CP47 to PSII RC leads to a release of PSII assembly factors such as HCF136 and in turn ensures the correct configuration of the C-terminal extension of pD1 at a specific spatial location for ready access of the CtpA enzyme[7,14,39]. In the *fpb1* and *pam68* mutants, incorporation of labelled methionine in pD1/D1 is reduced but the ribosome footprints on *psbA* mRNA is comparable to WT plants (Fig. 2b, 6b). The likely reason for this is that the newly synthesized pD1/D1 are rapidly degraded because of limited binding partners for further assembly into intact PSII complex. Indeed, a large portion of newly synthesized pD1 in *fpb1* and *pam68* is present in the region of unassembled proteins in our 2D BN/SDS-PAGE (Fig. 2f). In addition, the abnormal ribosome pausing on the *psbA*-V may also influence the synthesis rate of pD1 in *fpb1* and *pam68* (Supplementary

Fig. 9 and Data 3). Based on these facts, we propose that reduced accumulation of newly synthesized D1 and retarded processing of pD1 in the *fpb1* and *pam68* mutant plants are secondary effects caused by limiting amounts of CP47 protein during the biogenesis of PSII.

FPB1 is an evolutionary conserved thylakoid protein in land plants and algae but is absent from the photosynthetic prokaryotes (Fig. 4a), suggesting that FPB1 represents a eukaryotic acquisition specifically facilitating PSII assembly. By contrast, PAM68 is conserved in all photosynthetic organisms[12]. Absence of PAM68 in Arabidopsis leads to a severe decrease of PSII accumulation, but a normal level of PSII was observed in the *Synechocystis* strain *insO933*, in which the PAM68 ortholog (Sll0933) is deleted[12]. Even so, the results of co-purification of Sll0933 with CP47 suggest a conserved role for the PAM68 ortholog in CP47 biogenesis in cyanobacteria and in lands plants[17]. However, the minor effect on PSII biogenesis in *insO933* may suggest that the PAM68/FPB1-mediated process during CP47 biogenesis is less critical in cyanobacteria than in land plants. Absence of FPB1 in cyanobacteria may also support this scenario. Most likely acquisition of FPB1 in chloroplasts during evolution may have enhanced the efficiency of PAM68-mediated CP47 biogenesis or overcome some shortcomings in the cyanobacterial PAM68-mediated pathway to adapt to the environment and needs of chloroplasts in land plants. Alternatively, FPB1 plays different but closely related roles with PAM68 during the biogenesis of CP47.

How do FPB1 and PAM68 assist CP47 biogenesis? Five major peaks of ribosome footprints along the *psbB* ORF were detected, suggesting that they correspond to ribosome pausing positions (Fig. 6c–e). Previous observations indicate that the synthesis of nascent integral thylakoid proteins occurs in the chloroplast stroma[18]. It has been shown that cpSRP54 binds to the ribosome through its interaction with the uL4c ribosomal subunit and docks the RNC to the translocon by interaction with FtsY[20,40]. Since cpSRP54 and FtsY were co-purified with FPB1 and interact with FPB1/PAM68 in our BiFC assay (Supplementary Fig. 6a; Data 1), we propose that cpSRP54 binds to the ribosome and the N-terminal segment of CP47 that just emerges from the ribosome tunnel in the stroma and then the RNC-SRP complex is recruited to the translocon on thylakoids via the FtsY receptor (Step 1, Fig. 7). The positions of the second and third peaks correspond roughly to the 2nd and 4th TMDs of CP47, suggesting that ribosomes pause after these TMD segments emerge from the ribosomal tunnel during translation (Fig. 6c–e). The structure of mature CP47 revealed that TMDs of I/II and III/IV form a pair of closely interacting α-helixes with a short luminal loop, respectively (Supplementary Fig. 1a)[14]. Based on the model of co-translational insertion proposed recently[41], it is reasonable to speculate that these two pairs of α-helixes co-insert into thylakoids via the translocon and Alb3 systems (Steps 2 and 3, Fig. 7). Pausing of ribosomes during the translation of the 2nd and 4th TMD segments may provide enough time or space for the incorporation of pigments and other cofactors into these TMD.

The TMDs of V/VI also form a pair of closely interacting α-helixes but with a large luminal loop (E loop) in the mature CP47 protein (Supplementary Fig. 1a)[14]. Thus, its incorporation into thylakoids seems more challenging than that of the other two pairs of α-helixes. The fourth and fifth peaks correspond to the positions after the 5th and 6th TMDs of CP47 emerging from the ribosomal tunnel, suggesting ribosomes pause at these positions (Fig. 6c–e; Steps 4 to 6 in Fig. 7). At least two alternative possibilities need to be considered for the incorporation of this part of CP47. The first one is that the TMD V, E loop, and TMD VI are sequentially inserted into thylakoids (Steps 5 and 6, Fig. 7). Alternatively, after emerging from the ribosomal tunnel, TMD V stays along the stromal side of thylakoids and waits for TMD VI. The E-loop is translocated into the thylakoid lumen before TMDs V/VI (Steps 5′ and 6′, Fig. 7). Since increased ribosome stalling at the 6th TMD of CP47 was detected in the *fpb1* and *pam68* mutants (Fig. 6c–e), FPB1 and PAM68 may

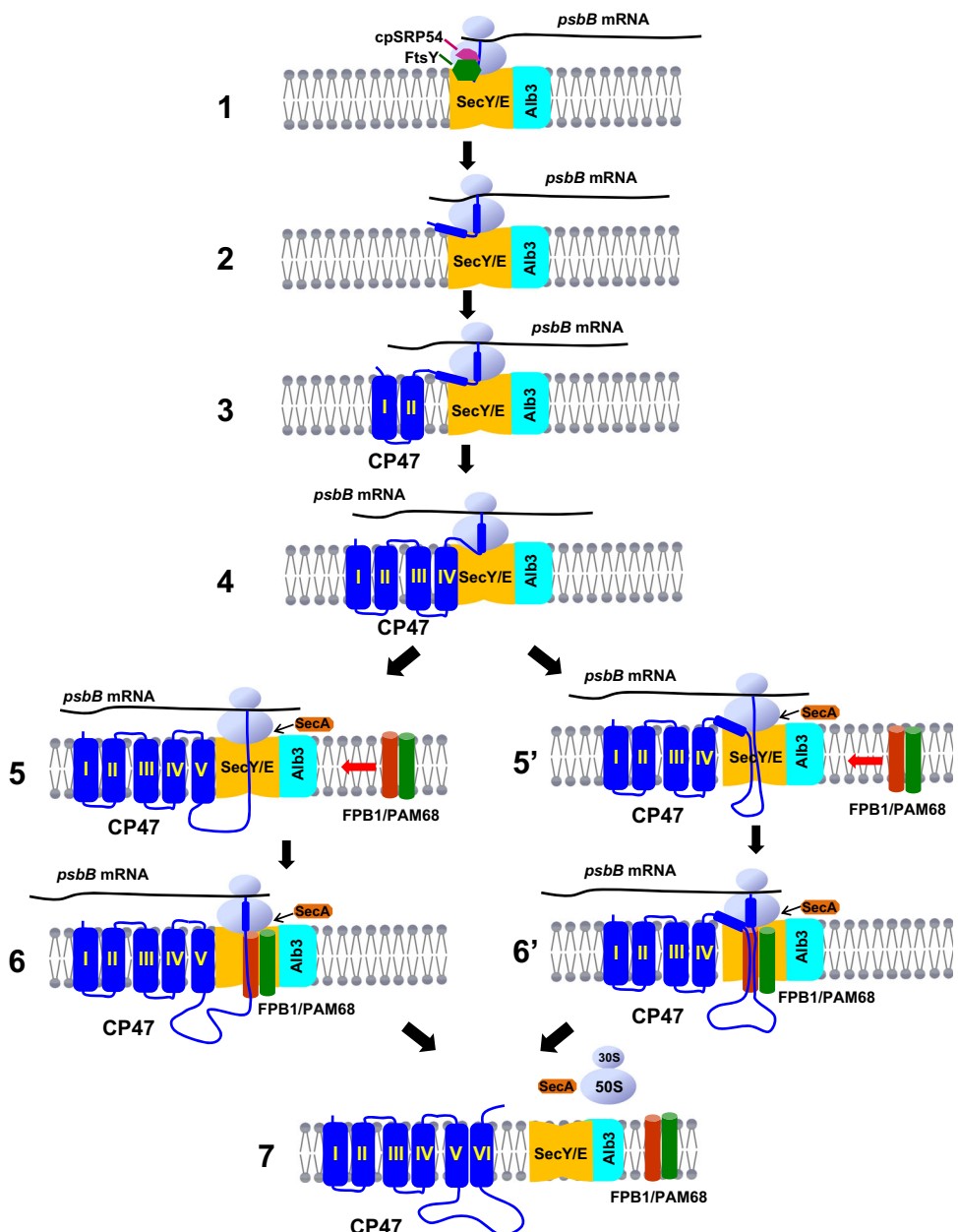

**Fig. 7 | A possible model for the co-translational assembly of CP47 and the function of FPB1 and PAM68 in chloroplasts.** After translation initiation in the chloroplast stroma, the ribosome moves along the *psbB* mRNA transcript. Once the first transmembrane segment of CP47 emerges from the ribosome exit tunnel, cpSRP54 binds to the RNC and docks the ribosomes to the SEC/Alb3 system via the interaction with FtsY (Step 1). TMDs of I/II and III/IV are co-translationally inserted into thylakoids sequentially (Steps 2 to 4). In the later process, the TMD V, E loop, and TMD VI are sequentially inserted into the thylakoid membrane (Steps 5 and 6) or TMDs V/VI are translocated together following the E loop (Steps 5′ and 6′). These two steps require the participation of FPB1 and PAM68 to facilitate the efficient insertion or folding of CP47. The cpSecA protein may be involved in the translocation of the large E loop with SecY/E. The facilitator proteins are released once the biogenesis of CP47 is completed (Step 7).

facilitate the insertion of this TMD (Steps 6) or the last two TMDs (Steps 5′ and 6′) into the SEC and Alb3 systems and promote the subsequent interaction between CP47 and its partners in thylakoids. In bacteria, SecA, an ATPase motor, is required for translocation of the inner membrane proteins with a large periplasmic loop (greater than 30 aa) connecting TMDs or single transmembrane inner membrane proteins with periplasmic loops[42,43]. Recently, the SecA ATPase cycle was shown to be required for the regulation of SecY/E opening and closing[44]. Interaction between cpSecA and FPB1/PAM68 in our BiFC assays and copurification of cpSecA with FPB1 raise the possibility that cpSecA mediates the translocation of the E loop of CP47 (Fig. 7; Supplementary Fig. 6 and Data 1).

Based on the co-translational assembly model (Fig. 7)[18,19,41], we propose that FPB1 and PAM68 act as molecular chaperones facilitating the incorporation of CP47 into thylakoids during its co-translational assembly process. However, the translation of a thylakoid protein with multiple TMDs and its subsequent insertion into thylakoids are almost synchronous and tightly connected. We could not unambiguously distinguish if FPB1 and PAM68 facilitate primarily *psbB* translation elongation through the binding of the nascent CP47 peptide or if FPB1 and PAM68 are primarily involved in CP47 insertion into thylakoids, which in turn feedback-regulates translation elongation of *psbB*. It is also possible that FPB1 and PAM68 participate in the two connected processes simultaneously during the biogenesis of CP47.

We cannot exclude another possible model in which both FPB1 and PAM68 could facilitate folding of CP47, especially TMDs V/VI and the E-loop, perhaps together with Alb3. Although translational pausing is often intrinsically robust[45], misfolded CP47 would likely be degraded by SEC-associated proteases before translation is completed which would further perturb ribosomal activity. In *Synechocystis*, chlorophyll *a* and β-carotene molecules were found in the non-assembled CP43 and CP47 proteins and the last enzyme for Chl biosynthesis was shown to interact with the YidC insertase[46,47]. Thus, pigment binding to CP47 may occur cotranslationally during CP47 biogenesis[5]. In maize, however, deficiency of chlorophyll does not alter the pausing at specific sites of CP47[48], which is in line with the possibility that misfolding or degradation of CP47 affects the activity of *psbB* mRNA-associated ribosomes to a limited extent. In the Synechocystis *pam68* mutant, the synthesis of CP47 was restored by upregulated chlorophyll biosynthesis and the pigments were present in the PAM68-CP47 subcomplex of *Synechocystis*[17]. This implies that PAM68 is also involved in facilitating the insertion of chlorophyll into CP47 in cyanobacteria. Whether FPB1 and PAM68 facilitate the insertion of chlorophyll into CP47 in chloroplasts remains to be elucidated.

In summary, in this work, we have identified and characterized a PSII biogenesis factor FPB1 which was acquired by photosynthetic eukaryotes during evolution. We provide genetic and biochemical evidence that FPB1 interacts with the previously identified assembly factor PAM68 and they synergistically orchestrate CP47 biogenesis during PSII assembly. CP47 was proposed to be co-translationally inserted into thylakoid membrane. By interacting with the SecY/E translocon machinery and Alb3 integrase, FPB1 and PAM68 are likely to facilitate the co-translational integration of the last two TMDs and the connecting loop of CP47 into thylakoids and subsequently into the PSII core complex.

## Methods

### Plant material and growth conditions

Mutants *fpb1* (SALK_048033), *lpa1* (GABI_655D01), and *alb3* (SALK_070924) were obtained from NASC. The T-DNA insertion mutant *pam68* was isolated from sets of T-DNA insertion pools (Stock No.: CS31400) and the pSKI015 was inserted into the first intron of *PAM68*, which confirmed by sequencing the PCR products. The seeds of *hcf107* (hcf107-2), *hcf136*, and *ctpa* (atctpa-1) were kindly provided by Prof. Jörg Meurer or Aigen Fu, respectively[7,15]. Mutants of *hcf173* (GABI_246C02) and *hcf244* (GABI_088C04) were used as described[27].

*Arabidopsis thaliana* (Columbia, Col-0) plants wild type, *fpb1*, *pam68*, and *lpa1* were grown in soil and other mutants were grown in MS medium containing 3% sucrose under a 12-h photoperiod with an irradiance of ~40–50 µmol photons m$^{-2}$ s$^{-1}$ in the greenhouse at 23 °C. Low light of ~5 (Fig. 3c) or 20–30 (Fig. 3a, b) µmol photons m$^{-2}$ s$^{-1}$ was used for growth of the *fpb1 pam68* double mutant in the same greenhouse. For complementation of the *fpb1* mutant, genomic DNA fragments of wild-type *At3g51510* were cloned into the pBIN19 vector and then transferred into *Agrobacterium tumefaciens* C58C. The bacteria were used to transform *fpb1* plants by the floral dipping method. The primers used in this work are listed in Supplementary Table 1.

### Chlorophyll fluorescence analysis and P700 oxidation measurements

Images of chlorophyll fluorescence were captured by using the MAXI version of the IMAGING-PAM system with default parameters (Walz, Effeltrich, Germany). Four-week-old plants were dark adapted for 30 min and a blue measuring light (1 Hz, intensity 2) was applied to determine the minimal fluorescence ($F_o$). Then, a saturating light pulse (blue light, intensity 10) was used to measure the maximal fluorescence (Fm) and Fv/Fm (($Fm-F_o$)/Fm) was calculated and displayed on a false colour scale. Induction curves of chlorophyll fluorescence were recorded using the PAM2500 system (Walz, Effeltrich, Germany).

Before measurements, the leaves were kept in darkness for 30 min to open PSII reaction centres. A weak measuring light (red light, intensity 2) and a saturating pulse (red light, intensity 10) were sequentially applied to determine $F_o$ and Fm, respectively. Then, the steady-state fluorescence was recorded with actinic light (AL, 64 µmol photons m$^{-2}$ s$^{-1}$) for 4 min.

P700 absorbance changes were measured with a Dual-PAM-100 (Walz, Effeltrich, Germany). The leaves of 4-week-old plants were dark-adapted overnight and then illuminated with AL light (red light, intensity 8) for 3 min. Absorbance changes of P700 at 830 nm were recorded after turning off the AL light (ΔA, representing the oxidised P700 under AL light conditions). Then, a far-red (FR) light (intensity 10) was applied to induce the oxidation of total P700 (ΔAmax). During the illumination with FR, a strong saturating light pulse (red light, intensity 10) was applied to excite PSII and induce electron flow from PSII to PSI, resulting in a re-reduction of P700$^+$.

### RNA gel blot and polysome association analyses

RNA gel blot analysis was performed basically according to our previous study[49]. Total RNA was isolated from the rosette leaves of 4-week-old plants using the Trizol reagent (Thermo Fisher Scientific). Total RNA (5 µg for *psbB*, *psbD*, and *psbKI*; 2.5 µg for *psbA*, *psbC*, *psbEFLJ*, and *ACTIN7*) was separated on a 1.5% (w/v) formaldehyde agarose gel and then capillary blotted onto nylon membranes (Hybond-N$^+$, GE Healthcare). Hybridization was performed at 55 °C using DIG Easy Hyb buffer (Roche) and the signal was visualized with a LuminoGraph WSE-6100 (ATTO). RNA Marker RL6000 (Takara) was used as a marker.

Polysome association analysis was performed basically according to Barkan[50]. Three-week-old plants were moved to the laboratory from the greenhouse and then the leaves were ground in liquid nitrogen and then homogenized in a polysome extraction buffer (0.2 M Tris-HCl, pH 9.0, 0.2 M KCl, 35 mM MgCl$_2$, 25 mM EGTA, 0.2 M Sucrose, 1% [v/v] Triton X-100, 2% [v/v] polyoxyethylene-10-tridecyl ether, 0.5 mg/mL heparin, 100 mM β-mercaptoethanol, 100 µg/mL chloramphenicol, and 25 µg/mL cycloheximide). After centrifugation at 20,000 g at 4 °C for 5 min, sodium deoxycholate were added to the supernatants at a final concentration of 0.5% (w/v) to solubilize the microsomal membranes. Samples were centrifuged at 20,000 g for 15 min at 4 °C and the supernatants were layered on top of continuous sucrose gradients (15–55% sucrose in 40 mM Tris-HCl, pH 8.0, 20 mM KCl, 10 mM MgCl$_2$, 100 µg/mL chloramphenicol, 0.5 mg/mL heparin, and 25 µg/mL cycloheximide). The density gradients were centrifuged (240,000 g) for 65 min at 4 °C in the SW Ti55 rotors. The gradients were equally divided into 10 fractions and RNA was isolated for RNA gel-blot analyses as described above. To disrupt polysome association with mRNA, the leaves were homogenized in polysome extraction buffer (the concentration of KCl is increased to 0.5 M) containing 500 µg/mL puromycin. After centrifugation at 20,000 g at 4 °C for 5 min, the supernatants were incubated at 37 °C for 10 min and then 0.5% sodium deoxycholate was added according to Barkan[51].

### In vivo labelling of chloroplast proteins

In vivo protein labelling was performed as described[28,49]. Primary leaves of 12-d-old young seedlings were pre-incubated in 20 µg/mL cycloheximide for 30 min to inhibit cytosolic protein synthesis and then radiolabeled with 1 mCi/mL [$^{35}$S]-Met (specific activity >1000 Ci/mmol; Perkin Elmer Life Sciences, Boston, MA) in the presence cycloheximide for 20 min at 23 °C. To investigate the fate of the labelled thylakoid proteins, 1 mM unlabelled Met was used for the subsequent chase following the pulse labelling. After labelling and chase, thylakoids were isolated and subsequent 2D BN/SDS-urea-PAGE, 12.5% SDS-urea-PAGE or 16% Tricine-SDS-PAGE were used to separate thylakoid protein complexes and individual subunits, respectively.

## Total Leaf Protein and Thylakoid Preparation, BN-PAGE, and Immunoblot Analysis

For total leaf protein, the leaves of four-week-old were ground in a buffer containing 125 mM Tris-HCl pH 8.8, 1% (w/v) SDS, 10% (v/v) glycerol, 50 mM $Na_2S_2O_5$ with a conical plastic in an eppendorf tube. After incubation for 5 min at room temperature, solubilized proteins were separated by centrifugation at 13,000 g for 10 min at 4 °C. The protein concentration was determined with a BioRad Dc Protein Assay kit (Bio-Rad, USA). Thylakoid membrane isolation was performed according to ref. [49]. Briefly, leaves of four-week-old plants were homogenized in a medium containing 0.33 M sorbitol, 30 mM Tricine-KOH, pH 8.4, 5 mM EGTA, 5 mM EDTA, and 10 mM $NaHCO_3$. After centrifugation for 5 min at 4200 g, the pellet was resuspended in 0.3 M sorbitol, 20 mM Hepes-KOH, pH 7.6, 5 mM $MgCl_2$, and 2.5 mM EDTA and the chloroplasts were osmotically ruptured in the same buffer but lacking sorbitol. After centrifugation at 10,000 g for 5 min at 4 °C, the pellet containing the thylakoid membranes was used for subsequent BN-PAGE or other experiments and the supernatant corresponding to the stromal proteins was subjected to a second round of centrifugation (15,000 g for 10 min at 4 °C). Protein contents of stromal and thylakoid samples were determined with a Protein Assay Kit (Bio-Rad, USA) and a BioRad Dc Protein Assay kit (Bio-Rad, USA), respectively. BN-PAGE, SDS-urea-PAGE, and immunoblot analysis were performed as described[49].

## Salt Washes and Trypsin Treatment of Thylakoids

Thylakoid membranes isolated from wild-type plants were resuspended in 0.3 M sorbitol, 20 mM Hepes-KOH, pH 7.6, 5 mM $MgCl_2$, and 2.5 mM EDTA at a chlorophyll concentration of 0.5 mg mL$^{-1}$ and treated with 0.25 M NaCl, 0.2 M $Na_2CO_3$, 1 M $CaCl_2$, or 0.1 M NaOH with general mixing. After incubation on ice for 30 min, membranes were pelleted by centrifugation for 10 min at 20,000 g and 4 °C. The proteins released from thylakoids by salts were precipitated from the supernatant with 80% acetone and immunoblot analysis was performed using specific antibodies.

Thylakoids were resuspended in a buffer containing 0.3 M sorbitol, 20 mM Hepes-KOH, pH 7.6, 5 mM $MgCl_2$, and 2.5 mM EDTA at a chlorophyll concentration of 0.1 mg mL$^{-1}$ and 10 µg mL$^{-1}$ of trypsin was added. After incubation on ice for 5, 10, and 20 min, thylakoids were precipitated by centrifugation for 5 min at 20,000 g at 4 °C, and proteins were denatured immediately with SDS-PAGE buffer containing 1 mM PMSF.

## Antibody generation and resources

Polyclonal antibodies against purified recombinant FPB1 (amino acid residues 37–101) and Alb3 (amino acid residues 340–462) were generated by PhytoAB company (USA) in rabbits. Antibodies against pD1 (9 amino acid residues in the C terminus of pD1, 1:5000) and CtpA (1:1000) were obtained from Prof. Aigen Fu (Northwest University, China). For Co-IP, polyclonal antibodies against synthetic peptide of FPB1 (EAANKEAQEQDKRDGFC), PAM68 (PHY2289A) and YCF4 (PHY1363A) antibodies used for Co-IP were generated or obtained from PhytoAB (USA). Further information on other antibodies is available in the Nature Research Reporting Summary linked to this article.

## Co-immunoprecipitation

Affinity-purified pre-antiserum, FPB1, PAM68, and YCF4 antibodies were coupled to the CNBr-activated agarose at a ratio of 2 mg protein per 1 mL agarose (Sangon Biotech). Wild-type thylakoids (1 mg chlorophyll mL$^{-1}$) were solubilized using 1% dodecyl-β-D-maltoside (DM) for 30 min on ice in the co-IP buffer containing 50 mM Tris-HCl, pH 8.0, 100 mM NaCl, 1 mM EDTA, 0.1% gepal CA-630 (Sigma-Aldrich), and protease inhibitor (cOmplete, Roche). Non-solubilized membranes were removed by centrifugation at 12,000 g for 10 min and the supernatant containing 2.5 mg chlorophyll of solubilized membrane proteins was diluted 5 times with co-IP buffer and incubated with 100 µL antibody-coupled CNBr-activated agarose overnight at 4 °C. The beads were washed seven times with 1 mL co-IP buffer and one time with 1 mL washing buffer (20 mM Tris-HCl, pH7.5). The bound proteins were eluted with 60 µL of 1.5×Laemmli buffer (90 mM Tris-HCl, pH 6.8, 3% SDS, 10% glycerol, and 0.01% bromophenol blue) at 37 °C for 30 min. The eluted proteins were used for immunobloting or mass spectrometry analysis.

For crosslinking experiments, thylakoids were crosslinked with 1.5 mM DSP (Dithiobis (succinimidyl propionate)) for 2 h on ice and then stopped by adding 50 mM Tris-HCl, pH7.5. Thylakoids were solubilized in RIPA buffer (1% TritonX-100, 1% sodium deoxycholate, 0.1% SDS, 0.15 M NaCl, 0.01 M sodium phosphate, pH 7.2) and then co-immunoprecipitated with CNBr-activated agrose coupled antibodies. After incubation at 4 °C overnight, the bound proteins were washed 7 times with RIPA buffer and one time with 1 mL washing buffer (20 mM Tris-HCl, pH7.5) and then eluted in SDS-sample buffer (without β-mercaptoethanol) at 37 °C for 30 min. Proteins were incubated with 6% β-mercaptoethanol to open the bonds crosslinked by DSP before loading samples onto the SDS-PAGE gels.

## Split-ubiquitin yeast two hybrid assay

The sequence encoding mature FPB1 (amino acids 37 to 181) was cloned into the vector pNCW (Dualsystems Biotech) used to construct the bait plasmids, in which the Cub-LexA-VP16 module was fused to the N-terminus of FPB1 (Cub-FPB1). Sequences encoding mature Alb3 and SecY1 were cloned into the vector pCCW-SUC (Dualsystems Biotech) to construct the bait plasmids, in which the Cub-LexA-VP16 module was fused to the C-terminus of Alb3 (Alb3-Cub) and SecY1 (SecY1-Cub). For the prey vectors, coding sequences of mature thylakoid proteins whose N termini are exposed to the chloroplast stroma (CP47, CP43, PsaA, LPA1, LPA2, FPB1, and PAM68) were cloned into the pDSL-Nx vector (Dualsystems Biotech) and YCF4 was cloned into the pDSL-2xN vector to ensure that the fused NubG (NubG-X, X represents prey proteins) is present in the cytoplasm of yeast. The plasmids pAI-Alg5 encoding Alg5-NubI and pDL2-Alg5 expressing Alg5-NubG were used as negative and positive controls, respectively. Bait and prey vectors were co-transformed into yeast strain NMY32 and positive colonies were selected on synthetic medium lacking Leu and Trp (-LT). Interactions were verified by growing yeast clones on selective medium (SD-leu-trp-his-ade, -TLHA) plats for 2 to 4 days at 28 °C.

## Bimolecular fluorescence complementation (BiFC) assay

Because of the topologies of proteins in thylakoids, N- and C-terminal part of YFP (YFP$^N$ and YFP$^C$) fused to the two interacting partners must be located on the same side of thylakoids in the BiFC assay. Fragments containing the coding sequences of FPB1, PAM68, and YCF4 were cloned in pUC-SPYNE vector to fuse the YFP$^N$ at their C-terminal end (FPB1-YFP$^N$, PAM68-YFP$^N$, and YCF4-YFP$^N$, respectively)[52]. On the other hand, fragments containing the coding sequences of interacting partners (FPB1, PAM68, Alb3, cpSRP54, FtsY, LPA1, LPA2, SecA1, and YCF4) were cloned in the pUC-SPYCE vector to express X-YFP$^C$ (X represents interacting partners), since their C terminus or entire protein are present on the stromal side of thylakoids. For the proteins with their N terminus in the stroma side of thylakoids, the pUC-SPYCE vector was modified to generate pUC-RbcS-TP-SPYCE vector, which harbours targeting sequences of RbcS (RbcS-TP) to transport the fusion proteins into chloroplasts. Fragments encoding the mature proteins (SecY1 and SecE1) were cloned in the pUC-RbcS-TP-SPYCE vector to express RbcS-TP-YFP$^C$-X fusion proteins (X represents interacting partners). Plasmids expressing YFP$^N$ and YFP$^C$ fusion proteins were cotransformed into Arabidopsis protoplasts. The YFP fluorescence was observed using a confocal microscope (FV3000, Olympus).

## Ribosome profiling and RNA-seq

Ribosomal profiling and RNA-seq as well as the data analysis were performed by Gene Denovo Biotecholgy CO., (Guangzhou China) according to Ingolia et al.[33] with a few modifications[28]. Leaves of four-week-old Arabidopsis plants (0.4 g) were frozen in liquid nitrogen and ground to powder with a mortar and pestle, and then dissolved in 400 μL of lysis buffer (20 mM Tris HCl pH 8.0, 140 mM KCl, 1.5 mM $MgCl_2$, 100 μg mL$^{-1}$ chloramphenicol, 100 μg mL$^{-1}$ cycloheximide, 1% Triton-X-100). After centrifugation at 20,000 g for 10 min at 4 °C, 300 μL supernatant was collected. To degrade unprotected RNA and DNA, a total of 7.5 μL RNase I (NEB, USA) and 5 μL DNase I (NEB, USA) were added to the lysate and gently mixed on a Nutator mixer at room temperature for 45 min. Then, 10 μL RNase inhibitor (SUPERase•In; Ambion, USA) was added to stop the nuclease digestion. During the digestion, size exclusion columns (Illustra MicroSpin S-400 HR Columns; GE Healthcare, USA) were equilibrated with 3 mL polysome buffer (20 mM Tris-HCl, pH 8.0, 100 mM KCl, 5 mM $MgCl_2$, 1% Nonidet P-40) by gravity flow and then centrifuged at 600 g at room temperature for 4 min. A total of 100 μL of digested lysate were added to the equilibrated columns and centrifuged at 600 g for 2 min. The elution sample was incubated with 10 μL 10% SDS and then the ribosome footprints (>17 nt) was isolated by using the RNA Clean and Concentrator-25 kit (Zymo Research, China). Next, rRNA was removed as reported previously[53]. Antisense DNA probes complementary to rRNA sequences were added. To digest rRNA and residual DNA probes, RNase H (NEB, USA) and DNase I was added. Then, ribosome footprints were purified using magnetic beads (Vazyme, China).

Ribo-seq libraries were prepared using NEBNext Multiple Small RNA Library Prep Set for Illumina (NEB, USA). Firstly, adaptors were added to both ends of ribosome footprint RNA. After reverse transcription and PCR amplification, DNA fragments with sizes at 140–160 bp (representing insert sizes of 20–40 bp) were enriched to generate Ribo-seq libraries, which was further sequenced using Illumina HiSeqTM X10 (Gene Denovo Biotechnology Co., China) to a depth of 50–84 million reads.

For RNA-Seq, total RNA was isolated by using Trizol reagent kit (Invitrogen, USA). The mRNA was enriched by removing rRNA with the Ribo-Zero Magentic kit (Epicentre, USA) and fragmented into short fragments. First-strand cDNA were transcribed with random primers. Second-strand cDNA was synthesized and then purified using the QiaQuick RPC extraction kit (Qiagen, Netherlands). The synthesized cDNA was end repaired, poly(A) added, and ligated to Illumina sequencing adaptors (NEB, USA). The libraries were prepared and sequenced using Illumina HiSeqTM X10 to a depth of 40–47 million reads.

## Ribo-Seq and RNA-Seq data analysis

Raw reads containing over 50% of low quality bases or over 10% of N bases were removed. Adaptor sequences were trimmed using fastP. Short reads alignment tool Bowtie2[54] was used for mapping reads to ribosomal RNA (rRNA) database, GenBank, Rfam database, and miRBase. The reads mapped to rRNAs, transfer RNAs (tRNA), small nuclear RNAs (snRNA), small nucleolar RNAs (snoRNA), and miRNA were removed and the remaining reads were assigned to different genomic features (5'UTR, CDS, 3'UTR and intron) based on the position of the 5' end of the alignment as described[55]. Reads number in the open reading frame of coding genes was calculated by software RiboTaper[56] and the gene expression level was normalized by using FPKM (Fragments Per Kilobase of transcript per Million mapped reads) method.

## LC-MS/MS analysis

LC-MS/MS analysis was used for protein identification of immunoprecipitation. 20 μL eluted samples of immunoprecipitation were separated by 12.5% SDS-PAGE and each gel lane was cut in consecutive gel slices followed by destaining twice using $NH_4HCO_3$/ACN. Proteins were digested in-gel with DTT, IAA and trypsin. Digested peptides in the gel slices were then desalted using StageTips with C18 Cartridge (Sigma). Peptides extracts for each gel slice were analysed by High Performance Liquid Chromatography (HPLC) using EASY column (Fisher Scientific, USA) coupled with a Q-Exactive mass spectrometer (Thermo Finnigan). Resulting MS/MS spectra were searched against the predicted Arabidopsis proteome (35386 entries, 20191018) with maxquant software (version 2.3).

## Statistics and reproducibility

Data analyses were done with Microsoft Excel (2016) and graphs were generated with Origin 6.0 or GraphPad Prism (version 10.1.2) software. Values are indicated as mean ± standard deviations (SD). The biological replicates of experiments presented in this study are indicated in the respective figure legends. No data were excluded from the analyses and no statistical method was used to predetermine sample size. The experiments were not randomized and the investigators were not blinded to allocation during experiments and outcome assessment.

## Reporting summary

Further information on research design is available in the Nature Portfolio Reporting Summary linked to this article.

## Data availability

All data supporting the findings of this study are available in the paper and the Supplementary Information. All other data that support the findings of this study are available from the corresponding author upon request. The raw data of RNA-Seq and ribosome profiling generated in this study have been deposited in the GenBank (NCBI) Sequence Read Archive (SRA) with accession number PRJNA948060. The MS raw data are available through the MassIVE repository with accession ID: MSV000094131 (https://massive.ucsd.edu/ProteoSAFe/dataset.jsp?task=c74dd58b934141f6a87b92f496983664). Source data are provided with this paper.

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

## Acknowledgements

We thank Prof. Aigen Fu for providing the pD1 and CtpA antibodies as well as Prof. Jörg Meurer and NASC for providing the mutant seeds. This work was supported by the Natural Science Foundation of Shanghai (No.22ZR1446000 to L.Z.), and the Shanghai Engineering Research Centre of Plant Germplasm Resources (No.17DZ2252700).

## Author contributions

L.P. conceived the study and designed the experiments. L.Z. and J.R. led the experimental work and together with F.G. and Q.X. performed most experiments. L.P.C. and L.C. performed BiFC assays. L.Z., J.R., Q.X., and L.C. analysed the RNA-Seq and Ribo-Seq data. Z.L., M.K., and H.M. helped with the protein labelling experiments. Data was analysed by L.Z., J.R., F.G., Q.X., L.P.C., L.C., J.D.R., and L.P. The paper was written by J.D.R. and L.P. with contribution from all other authors.

## Competing interests

The authors declare no competing interests.
