## [Peer Review File · Nature Communications]

REVIEWER COMMENTS

Reviewer #1 (Remarks to the Author):

In this work, Lin Zhang et al. characterized an Arabidopsis protein that was named FPB1 (Factor required for biogenesis PsbB1). Authors demonstrated that this protein is localized in thylakoids and is involved in the biogenesis of CP47 subunit of PSII. In the absence of FPB1, the synthesis and accumulation of CP47 is apparently impaired. The manuscript also shows nicely that the phenotype of *fpb1* mutant is similar to Arabidopsis mutant line lacking PAM68, which is a protein factor previously connected to the synthesis of CP47 in cyanobacteria. Indeed, the synthesis of CP47 in Arabidopsis *pam68/fpb1* double mutant is almost completely abolished. This part of the manuscript is novel, carefully performed and well interpreted.

In the second part of the manuscript, authors attempted to provide a mechanistic explanation of the FPB1/PAM68 function; however the obtained data are not so conclusive and the model lacks a solid support. Although it is likely that FPB1 and PAM68 perform a similar task (they seem to have even a similar structure) and are localized in the vicinity of ALB3, a clear-cut evidence for their physical interaction is missing. Mass spectrometry analysis of Co-IP samples indicates an enrichment of various auxiliary factors involved in biogenesis of PSII but these include also proteins most likely not involved in the biogenesis of CP47 (OHP2, HCF173, LPA1, HCF244). The proposed interaction with PSII intermediates containing OEC also does not make much sense as the attachment of OEC is a very late assembly step. The co-IP assays thus do not seem to be very specific. It would be useful to have additional control for co-IP/MS, e.g. any thylakoid membrane protein not related to biogenesis of PSII (b6f subunit, PSI subunit). It is even more important for the co-IP assay after crosslinking as this type of experiment is very prone to artefacts. Given the fact how weak is the signal of CP47 in immunoprecipitated samples, the conclusion that the FPB1 interacts with CP47 seems to be quite bold.

BiFC assay is popular in plants for protein-protein interaction studies but this method is a bit problematic regarding reliability and robustness. It is not just self-assembly of the YFP halves, but the investigated proteins can be massively over-accumulated, localized in wrong compartments, partially degraded, aggregated, not functional, etc. There are some recommended controls (e.g. Horstman A, *IJMS* 2014, 15: 9628). I do not know whether enough of protoplasts can be prepared for a simple SDS-PAGE followed by Western blot; however at least for the putative interactors important for the hypothesis (ALB3, PAM68), it would be interesting to: i) check the mass of fused proteins (FPB1-YFPn, ALB3-YFPc, PAM68-YFPc) to verify that these artificial proteins are intact and not fragmented, ii) compare the level of the native protein with the fused one.

Results of Ribo-seq analysis are interesting as other laboratories (e.g. Zoschke R, *Front Plant Sci*, 2017 8:385; Gawronski, *Plant Physiol.* 2018, 176:2557) did not report such well resolved ribosome pausing close to the positions of CP47 TM segments. However, the difference between WT, *fpb1* and *pam68* in

the distribution of ribosome footprints appears low even for the position of TM 6. In my opinion, the obtained Ribo-seq data are over-interpreted. Is it possible to quantify the difference?

Specific comments

The description of data in Supplemental Table I should be more detailed and I would suggest to sort the identified hits according to intensity or score rather than the coverage or number of peptides. In addition, rather than deliberately selected protein subsets, a full list of identified proteins should be shown in the table and sorted according to signal intensity or a score. Such a list will be probably long but it is important to know what are the most abundant hits in all samples including controls.

Fig. 2c – it is not clear how the signal of PsbH protein was identified. A number of proteins with mass similar to PsbH could be intensively labelled.

Why the ALB3 antibody was not included in the immunoanalysis of co-IP samples after crosslinking (Fig. 5b)?

The synthesis of CP47 in the *Synechocystis* PAM68 mutant was restored by upregulated chlorophyll biosynthesis (Bucinska, *L Plant Physiol*, 176: 2931–2942). How this result is consistent with the proposed role of FBP1 and PAM68 in the insertion of the last TM helix of CP47 (Fig. 7)? According to Zoschke R, (*Front Plant Sci*, 2017 8:385), the deficiency of chlorophyll does not alter the pausing at specific sites of CP47. How could high level of free chlorophyll in membranes improve the SecA-driven insertion of CP47 TM segments in the absence of FBP1/PAM68?

Reviewer #2 (Remarks to the Author):

In the present manuscript, Zhang et al. report on the molecular mechanism responsible for the co-translation insertion of CP47 subunit into the thylakoid membrane, an essential step at the basis of PSII biogenesis. This process requires the FBP1-PAM68 protein complex that acts in coordination with SECY/E and Alb3 thylakoid translocon complex. The manuscript presents very high-quality data obtained through a series of well organised and executed experiments. Through a mixture of genetics, biochemistry and molecular biology experiments, the authors were able to dissect the different molecular steps essential to integrate a six transmembrane-helix protein into the thylakoid membrane. The manuscript is written in an excellent way and all the different paragraphs are easy to read thanks to the very well organization and the high quality of the figures. From my point of view, the manuscript is fine as it is and does not deserve any improvement.

Reviewer #3 (Remarks to the Author):

The authors provide very exciting data for the novel PSII assembly factor FPB1, including its likely function in cotranslational CP47 maturation and assembly as well as the interaction with known factors acting in this pathway such as PAM68, SecY/E and ALB3. Most of their data are of excellent quality, very broadly characterize FPB1 and provide stimulating insights into the interconnected function of FPB1 and different known factors involved in PSII assembly. I have, however, a few major concerns and some minor points that should be addressed before publication:

Major points:

1. It would very challenging to design experiments that unambiguously distinguish if FPB1 facilitates primarily psbB translation (elongation) through binding the nascent CP47 peptide and the assembly defect is of secondary nature or if, vice versa, FPB1 is involved primarily in CP47 assembly, and the assembly defect feeds back on psbB translation (or if both of these hypotheses are true). Hence, I suggest to tone down some of the statements (e.g., in the abstract: “[...] involved in assisting the co-translational assembly of CP47, a subunit of the Photosystem II (PSII) core) and explain this dilemma in the discussion.
2. Line 154: As starting point of the results, it would be interesting for the readers and reviewers to understand how you found this mutant (was it a screen or similar thing)?
3. line 46 ff.: “Thus, our data demonstrate that, in coordination with the SecY/E translocon and the Alb3 integrase, FPB1 synergistically cooperates with PAM68 to facilitate the co-translational integration of the last two CP47 TMDs and the large loop between them into thylakoids and the PSII core complex.” – I think this statement is too strong. The data “strongly suggest or indicate” this conclusion.
4. Suppl. Figure 4: the polysome pattern for psbA looks unexpected – there are usually two peaks (bimodal distribution of psbA due to its large untranslated fraction; compare, for instance, to the hcf173 manuscript you cite). Similarly, the distribution of rRNAs looks unexpected – usually the major fraction is found in high sucrose density regions (compare for instance to Gao et al. 2022, Plant Cell). Do you have any explanation for this (e.g., developmental state used)?
5. line 281 ff.: “EDTA treatment induced clear shifts of psbB and psbEFJL transcripts towards lower molecular weight fractions in both the fpb1 mutant and WT (Fig. 2a), indicating that the psbB transcripts migrating deeper into the sucrose density gradient in fpb1 represent polysomes associated with mRNA” – most RNA-binding proteins would also require Mg²⁺ that is chelated by EDTA. Hence, disassembly by EDTA cannot unambiguously distinguish between ribosomal complexes and other large ribonucleoprotein particles. To do so a ribosome-specific disassembler such as puromycin would need to be used. Indeed, in line 445 ff. you state: “CP47 seems to be associated with high molecular weight unknown complexes which may represent protein aggregates” – if this is nascent CP47, it may also contain the psbB mRNA.

6. Fig. 5c: Interaction is found with almost every protein you tested, if I interpret the data right. How specific is this experiment? Your IP data with and without crosslinking seems to be much more informative. Please give some more information about the specificity and the interpretation of the BiFC assay data in the results. What is (are) your negative control(s)? Could you add negative controls?

7. Fig. 6: Two biological replicates should not been shown with standard deviation. I suggest to summarize the data shown in Fig. 6 (a-d) and Suppl. Fig. 7/8 for *fpb1* and *pam68* and show it with standard deviations (in the bar graphs as vertical lines, in the line graphs as shaded background) and add the *lpa1* data with single dots/ shades in different colors (representing the two performed experiments for bar and line graphs, respectively). This will possibly also show that the altered peak at the last TMD is an over-interpretation. For the bar plots it would be also good to show them with logarithmic y-axis scale to make easily visible what is up and what is down-regulated – this also then makes the extent of regulation equally visible in both directions (i.e., up and down).

8. Similarly, you state that “after normalization of ribosome footprint abundance, the fifth peak in *fpb1* and *pam68* is significantly higher than in WT in all three replicate samples (Figs. 6c and 6d, Supplementary Fig. 7b).” How was this significance statistically tested and is it the only found significant change in ribosome pausing between the different mutants and the WT throughout the whole chloroplast genome? A genome-wide analysis is needed here with statistical tests. I am afraid that you otherwise over-interpret the data at this point (although I have to admit that I am very much in favor of your hypothesis!).

9. Please also compare your WT *psbB* pausing sites to those in published datasets of Arabidopsis and other species for which Ribo-seq data are available (e.g., maize). If the peaks (i.e., pausing sites) are related to features of the nascent peptide (e.g., TMDs as you speculate) they should be conserved because the CP47 protein is highly conserved in the green lineage. Since the mRNA sequence is less conserved, mRNA features that lead to pausing may be less conserved in different species.

10. Figure 7: In their model step 1 ff. it would be good if there would be at least two ribosomes on the *psbB* mRNA. The downstream ribosome (with its nascent CP47) would localize the mRNA to the position where its translation is needed (close to the thylakoid membrane at PSII biogenesis sites), whereas the upstream ribosome is the one you show already (but the upstream ribosome needs to move from step 1 to step 2 to produce the nascent peptide that is bound by the targeting machinery). The way you show it now would represent the very first translation initiation on a *psbB* mRNA after or during transcription, which is not the most likely case to observe in chloroplasts – due to the long half-life of transcripts (it is more likely to see translation repeated translation initiation on a transcript that is already translated by further downstream ribosomes). Between step 4 and 5 make the arrow in a way that it connects the steps.

11. I did not find the information where the sequencing data is deposited.

Minor points:

1. Line 122 ff.” a total of 37 chloroplast-encoded proteins are intrinsic thylakoid proteins and 19 of them have been proposed to be co-translationally inserted into the membrane (Zoschke and Barkan, 2015)“ –

From my understanding the data in the cited paper demonstrate the co-translational insertion (not only “proposes” it).

2. Suppl. Figure 4: 1. It would be nice to label the size of the bands or, even better, provide ladder sizes so that readers (and reviewers) can decide which transcript isoform/s is/are shown. 2. It should be psbE/F/L/J.

3. line 441: circled instead of cycled.

4. line 674/75: “The large error bar with psbM mRNA is probably due to its small size resulting in a low number of RNA-Seq reads (Fig. 6a; Supplementary Table 2).” There are several other chloroplast-encoded small PSII subunits that do not show this high standard deviation... Could there be a biological reason for this behavior (e.g., related to the PsbM localization at the interphase of the PSII dimer)? You may consider to set a threshold that excludes genes with low read number to not distract the focus from your major points with high standard deviation of psbM.

5. line 694 ff.: “The ribosome footprints showed a peak at 32-33 nucleotides (Supplementary Fig. 6a), which is consistent with the cytosolic ribosome footprints in Maize and the 31-nucleotide modal size reported previously...” Usually for Arabidopsis smaller footprint sizes are reported – so your samples are probably slightly “under-digested”. I do not think that this influences the interpretation of the data at all but it makes the previous statement wrong.

6. line 697 ff.: “The frequency of footprints with a 5’ end at the adenine nucleotide was about 40% (Supplementary Fig. 6b), which is lower than the cytosol ribosome footprints reported in Maize (~50% in Chotewutmontri and Barkan, 2016).” I do not get the point – maybe explain a little bit better how you infer from the nature of the 5’ end nucleotide to the triperiodicity. Also maize (starting with lower case letter).

7. Supplemental Figure 7: if the number of replicates is below 3, no standard deviations should be shown. Subfigure c: state which set of gene was used for this meta-analysis – only cp genes or only nuclear genes or all genes, genes above a specific expression threshold, how many genes?

8. line 744: you refer to (Kim et al., 1991) which analyzes ribosome pausing in psbA but not psbB.

9. line 811 ff.: PsbH was also suggested to be involved in CES in Chlamy by Trösch et al. 2018.

10. Methods line 982 ff.: please make clear which material (including detailed growth conditions and developmental stage etc. pp.) was used for which experiment!

REVIEWER COMMENTS

Reviewer #1 (Remarks to the Author):

In this work, Lin Zhang et al. characterized an Arabidopsis protein that was named FPB1 (Factor required for biogenesis PsbB1). Authors demonstrated that this protein is localized in thylakoids and is involved in the biogenesis of CP47 subunit of PSII. In the absence of FPB1, the synthesis and accumulation of CP47 is apparently impaired. The manuscript also shows nicely that the phenotype of *fpb1* mutant is similar to Arabidopsis mutant line lacking PAM68, which is a protein factor previously connected to the synthesis of CP47 in cyanobacteria. Indeed, the synthesis of CP47 in Arabidopsis *pam68/fpb1* double mutant is almost completely abolished. This part of the manuscript is novel, carefully performed and well interpreted.

In the second part of the manuscript, authors attempted to provide a mechanistic explanation of the FPB1/PAM68 function; however the obtained data are not so conclusive and the model lacks a solid support. Although it is likely that FPB1 and PAM68 perform a similar task (they seem to have even a similar structure) and are localized in the vicinity of ALB3, a clear-cut evidence for their physical interaction is missing. Mass spectrometry analysis of Co-IP samples indicates an enrichment of various auxiliary factors involved in biogenesis of PSII but these include also proteins most likely not involved in the biogenesis of CP47 (OHP2, HCF173, LPA1, HCF244). The proposed interaction with PSII intermediates containing OEC also does not make much sense as the attachment of OEC is a very late assembly step. The co-IP assays thus do not seem to be very specific. It would be useful to have additional control for co-IP/MS, e.g. any thylakoid membrane protein not related to biogenesis of PSII (b6f subunit, PSI subunit). It is even more important for the co-IP assay after crosslinking as this type of experiment is very prone to artefacts. Given the fact how weak is the signal of CP47 in immunoprecipitated samples, the conclusion that the FPB1 interacts with CP47 seems to be quite bold.

---Thank you for your constructive comments. In the revised manuscript, we add YCF4 as an additional control for co-IP/MS (Fig. 5). YCF4 is a nonessential assembly factor for PSI and contains one transmembrane domain (Krech, et al., Plant Physiology, 2012). As shown in Fig. 5b and 5c, FPB1, PAM68, Alb3 as well as PSII subunits D1, D2, CP43, and CP47 cannot be co-purified with YCF4 using thylakoids or cross-linked thylakoids (Fig. 5b, c). We also performed mass spectrometric analyses of the YCF4-precipitated samples and the result is present in Supplementary Data 1.

Our co-IP assay using crosslinked thylakoid suggests physical interactions between FPB1 with PAM68, CP47 and Alb3 (Fig. 5b). In the revised manuscript we provide new evidence for their interactions using the split-ubiquitin yeast two-hybrid assays, which shows that FPB1 interacts with CP47, PAM68, LPA1, LPA2 but not

with CP43, PsaA and YCF4 (Fig. 6d). In addition, we found that Alb3 interacts with FPB1 and PAM68 and that SecY1 interacts with FPB1 and very weakly with PAM68 in yeast. Please refer to Fig. 5c and L579-L588.

BiFC assay is popular in plants for protein-protein Interaction studies but this method is a bit problematic regarding reliability and robustness. It is not just self-assembly of the YFP halves, but the investigated proteins can be massively over-accumulated, localized in wrong compartments, partially degraded, aggregated, not functional, etc. There are some recommended controls (e.g. Horstman A, IJMS 2014, 15: 9628). I do not know whether enough of protoplasts can be prepared for a simple SDS-PAGE followed by Western blot; however at least for the putative interactors important for the hypothesis (ALB3, PAM68), it would be interesting to: i) check the mass of fused proteins (FPB1-YFP^N, ALB3-YFP^C, PAM68-YFP^C) to verify that these artificial proteins are intact and not fragmented, ii) compare the level of the native protein with the fused one.

---In the revised manuscript, we added several controls and please refer to Supplementary Fig. 6a and 6b. The results showed that 1) No YFP signals were observed in protoplasts expressing FPB1-YFP^N/YCF4-YFP^C and PAM68-YFP^N/YCF4-YFP^C; 2) No YFP signals were observed in protoplasts expressing only YFP^N and various proteins fused with YFP^C; 3) No YFP signals were observed in protoplasts expressing YCF4-YFP^N and various proteins fused with YFP^C except for PYG7-YFP^C. According to your comments, we also performed western blot analysis using the protoplasts transformed with FPB1-YFP^N/PAM68-YFP^C and FPB1-YFP^N/Alb3-YFP^C (Supplementary Fig. 6c, d). The results showed that all three fusion proteins are intact and not fragmented. Although PAM68-YFP^C is likely over-accumulated in protoplasts, it cannot interact with YFP^N and YCF4-YFP^N.

Yes, we agree with this reviewer that BiFC assay is a bit problematic. However, our BiFC assay results are consistent with the results of Y2H, co-IP/MS, in which many PSII assembly factors and components of the translocon and integrase are co-purified with FPB1 and PAM68. These results may suggest that these proteins are directly interacted or, alternatively, are localized closely within a certain scaffold or area. We would like to move the BiFC results to the Supplementary Fig. 6 (L590-605).

Results of Ribo-seq analysis are interesting as other laboratories (e.g. Zoschke R, Front Plant Sci, 2017 8:385; Gawronski, P Plant Physiol. 2018, 176:2557) did not report such well resolved ribosome pausing close to the positions of CP47 TM segments. However, the difference between WT, *fpb1* and *pam68* in the distribution of ribosome footprints appears low even for the position of TM 6. In my opinion, the obtained Ribo-seq data are over-interpreted. Is it possible to quantify the difference?

---Thank you for your comments. According to your and Reviewer 3's comments, we compared the change in ribosome pausing between the different mutants and WT through the whole chloroplast genome. A threshold was set in this analysis, in which

the ribosome footprints reads at each position along the ORF less than 500 were not included in the assay (a total of 38 genes). The relative heights of the major peaks of the remaining 50 genes were calculated and the results are summarized in Supplementary Fig. 9 and Supplementary Data 3.

The genome-wide analysis showed that, for the relative height of the *psbB* fifth peak (*psbB*-V), the ratios of *fpb1* and *pam68* to WT are 1.310 ± 0.086 and 1.420 ± 0.041 , respectively. Only two peaks (*psbA*-V and *psbZ*-I) have higher relative heights in the *fpb1* and *pam68* mutants than the fifth peak of *psbB* (Supplementary Fig. 9). For the fifth peak of *psbA* (*psbA*-V), a higher relative height was observed in *fpb1*, *pam68*, *lpa1* as well as our previous reported *rbdl* mutant (Supplemental Dataset 5; Che et al., 2022). This peak is localized in the region of *psbA* mRNA encoding the third TMD, suggesting that the ribosomes pause during *psbA* translation when the third TMD of D1 emerges from the ribosomal tunnel. Interestingly, the rate of synthesis of D1 is reduced in all these four mutants (Fig. 2b; Peng et al., 2006; Che et al., 2022). It is possible that *psbA* translation at this position is secondarily feedback-regulated in the absence of FPB1 and PAM68. Alternatively, FPB1 and PAM68 are also directly involved in D1 synthesis and we cannot exclude this possibility. For the first peak of *psbZ* (*psbZ*-I), only *fpb1* had a relative higher height compared with WT.

In addition, the genome-wide analysis also detected several peaks with relative slightly higher heights (*psbD*-VI, *psaB*-I, *petA*-VI, *rps11*-I, and so on) or lower heights (such as *psbA*-III, *psbD*-1, *petB*-I, *atpH*-II, and so on) in the *fpb1* and *pam68* mutants compared with WT. This is possible due to the limitations of the Ribo-Seq technology but we cannot exclude the possibility that translation of these genes is directly or indirectly affected in these mutants. However, our genetic results and biochemical results indicate that *psbB* is the main target of FPB1 and PAM68. Thus, according to the genome-wide analysis, we would like to tone down some of the statements in the revised manuscript. Thanks to the reviewers' comments we believe that this revised part is more consistent and clearer. Please refer to L697-718.

Specific comments

The description of data in Supplemental Table I should be more detailed and I would suggest to sort the identified hits according to intensity or score rather than the coverage or number of peptides. In addition, rather than deliberately selected protein subsets, a full list of identified proteins should be shown in the table and sorted according to signal intensity or a score. Such a list will be probably long but it is important to know what are the most abundant hits in all samples including controls.

---We sorted the identified proteins according to their intensity in Supplementary Data 1 in the revised manuscript.

Fig. 2c – it is not clear how the signal of PsbH protein was identified. A number of proteins with mass similar to PsbH could be intensively labelled.

---We cited a previous report on the *hcf107* mutant (Felder et al., 2001). In the *hcf107* mutant, the synthesis of PsbH is completely blocked due to the loss of *psbH*

RNAs. In vivo protein labeling showed that PsbH is clearly radiolabelled as an 8-kDa protein in WT but absent in *hcf107* (Felder et al., 2001), indicating that only PsbH can be labeled in this region. Our results also detected this band just below 10-kDa in WT, *fpb1* and *pam68* mutants (Fig. 2C). Please refer to the L283-L285.

Why the ALB3 antibody was not included in the immunoanalysis of co-IP samples after crosslinking (Fig. 5b)?

---The blots using Alb3 antibody is added in the revised manuscript. Alb3 was detected in the FPB1-IP but not in PAM68-IP samples after crosslinking (Fig. 5b). However, the interaction between PAM68 and Alb3 was found in the split-ubiquitin yeast two-hybrid assays (Fig. 5c). Their interaction was also reported previously (Armbruster et al., 2010). This implies that crosslinking between Alb3 and PAM68 may not be efficient and we discuss this possibility in the text. Please refer to L506-L509.

The synthesis of CP47 in the *Synechocystis* PAM68 mutant was restored by upregulated chlorophyll biosynthesis (Bucinska, L Plant Physiol, 176: 2931–2942). How this result is consistent with the proposed role of FPB1 and PAM68 in the insertion of the last TM helix of CP47 (Fig. 7)? According to Zoschke R, (Front Plant Sci, 2017 8:385), the deficiency of chlorophyll does not alter the pausing at specific sites of CP47. How could high level of free chlorophyll in membranes improve the SecA-driven insertion of CP47 TM segments in the absence of FPB1/PAM68?

---Regarding the restoration of CP47 synthesis by an increase of chlorophyll synthesis, the situation might be different between *Arabidopsis* and *Synechocystis*, since FBPI is missing in cyanobacteria. Increasing the amount of chlorophyll may partially substitute the absence of PAM68 if the latter is also involved in facilitating the insertion of chlorophyll into CP47 in cyanobacteria. In *Arabidopsis*, there is no evidence showing that FPB1 and PAM68 are involved in facilitating the insertion of chlorophyll into CP47. However, our results and several previous reports suggest that the different steps of PSII assembly occur close to each other within a putative PSII assembly/repair scaffold. Deficiency of chlorophyll does not alter the pausing at specific sites of CP47, implying that insertion of chlorophyll into CP47 and insertion of CP47 TM segments into thylakoids may occur in different steps. We cite these two papers and briefly discussed this possibility in the revised manuscript. Please refer to L905-L917.

Reviewer #2 (Remarks to the Author):

In the present manuscript, Zhang et al. report on the molecular mechanism responsible for the co-translation insertion of CP47 subunit into the thylakoid membrane, an essential step at the basis of PSII biogenesis. This process requires the FPB1-PAM68 protein complex that acts in coordination with SECY/E and Alb3

thylakoid translocon complex. The manuscript presents very high-quality data obtained through a series of well organised and executed experiments. Through a mixture of genetics, biochemistry and molecular biology experiments, the authors were able to dissect the different molecular steps essential to integrate a six transmembrane-helix protein into the thylakoid membrane. The manuscript is written in an excellent way and all the different paragraphs are easy to read thanks to the very well organization and the high quality of the figures. From my point of view, the manuscript is fine as it is and does not deserve any improvement

---Thank you very much for your review of our manuscript. According to the comments of other two reviewers, we added some new results and re-analyzed our Ribo-seq data, which significantly improved our article. We believe that these new results increase the significance and interest of our study in the field of PSII biogenesis.

Reviewer #3 (Remarks to the Author):

The authors provide very exciting data for the novel PSII assembly factor FPB1, including its likely function in cotranslational CP47 maturation and assembly as well as the interaction with known factors acting in this pathway such as PAM68, SecY/E and ALB3. Most of their data are of excellent quality, very broadly characterize FPB1 and provide stimulating insights into the interconnected function of FPB1 and different known factors involved in PSII assembly. I have, however, a few major concerns and some minor points that should be addressed before publication:

Major points:

1. It would be very challenging to design experiments that unambiguously distinguish if FPB1 facilitates primarily psbB translation (elongation) through binding the nascent CP47 peptide and the assembly defect is of secondary nature or if, vice versa, FPB1 is involved primarily in CP47 assembly, and the assembly defect feeds back on psbB translation (or if both of these hypotheses are true). Hence, I suggest to tone down some of the statements (e.g., in the abstract: “[...] involved in assisting the co-translational assembly of CP47, a subunit of the Photosystem II (PSII) core) and explain this dilemma in the discussion.

---Thank you for your constructive comments. Indeed, we agree with you that it is very difficult to distinguish which process is primarily affected in the *fpb1* and *pam68* mutants because the translation of CP47 and its subsequent insertion into thylakoids are almost synchronous and tightly connected. We toned down our statements in the revised manuscript and also explained this dilemma in the discussion. Please refer to L894-L904.

2. Line 154: As starting point of the results, it would be interesting for the readers and reviewers to understand how you found this mutant (was it a screen or similar thing)?

---Thank you for your comments. We collected more than 500 T-DNA insertion Arabidopsis mutants from NASC, in which the mutated genes encode chloroplast-localized proteins and we screened for mutants defective in the accumulation of thylakoid protein complexes. The mutant *fpb1* is one of the mutants with a clear phenotype. We included this information in the revised manuscript. Please refer to L126-130.

3. line 46 ff.: “Thus, our data demonstrate that, in coordination with the SecY/E translocon and the Alb3 integrase, FPB1 synergistically cooperates with PAM68 to facilitate the co-translational integration of the last two CP47 TMDs and the large loop between them into thylakoids and the PSII core complex.” – I think this statement is too strong. The data “strongly suggest or indicate” this conclusion.

---Yes, we toned down our statements throughout the revised manuscript. Please refer to L40 and other parts of manuscript.

4. Suppl. Figure 4: the polysome pattern for *psbA* looks unexpected – there are usually two peaks (bimodal distribution of *psbA* due to its large untranslated fraction; compare, for instance, to the *hcf173* manuscript you cite). Similarly, the distribution of rRNAs looks unexpected – usually the major fraction is found in high sucrose density regions (compare for instance to Gao et al. 2022, Plant Cell). Do you have any explanation for this (e.g., developmental state used)?

---Thank you for your comments. We were also puzzled for a long time why only the *psbA* distribution pattern in our polysome analysis looks unexpected compared with other laboratories. After reading the report of Chotewutmontri P and Barkan A (2018, PLoS Genet 14(8):e1007555.), we realized that the growth light intensity may be one of the key factors for this problem. We cultured our plants at ~40-50 $\mu\text{mol photons m}^{-2} \text{ s}^{-1}$ (16-h photoperiod), while the seedlings were grown at ~50-70 and ~60-80 $\mu\text{mol photons m}^{-2} \text{ s}^{-1}$ in the *hcf173* and *hcf244* papers, respectively (Schult et al., 2007; Link et al., 2012). The *pam68* mutant was grown under a higher light intensity (~180 $\mu\text{mol photons m}^{-2} \text{ s}^{-1}$, 14-h photoperiod) (Armbruster et al., 2010). The maize and tobacco plants are usually grown under light intensities of >200 $\mu\text{mol photons m}^{-2} \text{ s}^{-1}$. In addition, Chotewutmontri P and Barkan A reported that the distribution of *psbA* mRNA shifted toward lower molecular weight fractions after shifting to the dark for 1 h (2018, PLoS Genet 14(8):e1007555.). Before the polysome isolation, we had to move the plants from our green house to the laboratory. During this process, the plants were kept in darkness or under room light (this period is usually about half an hour), which may also influence the distribution pattern for *psbA*. We have described our growth conditions in detail in the revised manuscript. Please refer to L939-L943 and L960.

For the distribution of rRNAs, the major fraction of rRNAs in tobacco was found in high sucrose density regions (Gao et al., 2022 Plant Cell 34: 2056-2079). However, this distribution seems different in Arabidopsis (*hcf173*, *hcf244*, *pam68*) (Schult et al., 2007; Link et al., 2012; Armbruster et al., 2010) and maize (Chotewutmontri P and Barkan A (2018, PLoS Genet 14(8):e1007555.)). In these

reports, two peaks of rRNAs were observed in sucrose density gradients. We have inserted a new figure of rRNA staining in Supplementary Fig. 4b, which is similar with those published previously.

5. line 281 ff.: “EDTA treatment induced clear shifts of *psbB* and *psbEFJL* transcripts towards lower molecular weight fractions in both the *fpb1* mutant and WT (Fig. 2a), indicating that the *psbB* transcripts migrating deeper into the sucrose density gradient in *fpb1* represent polysomes associated with mRNA“ – most RNA-binding proteins would also require Mg^{2+} that is chelated by EDTA. Hence, disassembly by EDTA cannot unambiguously distinguish between ribosomal complexes and other large ribonucleoprotein particles. To do so a ribosome-specific disassembler such as puromycin would need to be used. Indeed, in line 445 ff. you state: “CP47 seems to be associated with high molecular weight unknown complexes which may represent protein aggregates” – if this is nascent CP47, it may also contain the *psbB* mRNA.

---Thank you for your comments. We have repeated this experiment using the ribosome-specific disassembler puromycin to dissociate ribosomes. Similar to EDTA treatment, puromycin also induced shifts of *psbB* and *psbEFJL* transcripts towards lower molecular weight fractions in both the *fpb1* mutant and WT (new Fig. 2a), indicating that the *psbB* transcripts migrating deeper into the sucrose density gradient in *fpb1* represent polysomes associated with mRNA. This result is also consistent with the increased ribosome footprints in the *fpb1* and *pam68* mutants compared with WT and *lpa1* plants (Fig. 6b). Please refer to Fig. 2 and L235-239.

6. Fig. 5c: Interaction is found with almost every protein you tested, if I interpret the data right. How specific is this experiment? Your IP data with and without crosslinking seems to be much more informative. Please give some more information about the specificity and the interpretation of the BiFC assay data in the results. What is (are) your negative control(s)? Could you add negative controls?

---According to your and Reviewer 1’s comments, we added YCF4 as negative control in BiFC and Co-IP experiments. In addition, we also performed split-ubiquitin yeast two-hybrid assays, which show that FPB1 interacts with CP47, PAM68, LPA1, LPA2 but not with CP43, PsaA and YCF4 (Fig. 6c). In addition, we found that Alb3 interacts with FPB1 and PAM68 and that SecY1 interacts with FPB1 and very weakly with PAM68 in yeast. Please refer to new Fig. 5 and Supplementary Fig. 6 and L566-L605.

7. Fig. 6: Two biological replicates should not been shown with standard deviation. I suggest to summarize the data shown in Fig. 6 (a-d) and Suppl. Fig. 7/8 for *fpb1* and *pam68* and show it with standard deviations (in the bar graphs as vertical lines, in the line graphs as shaded background) and add the *lpa1* data with single dots/ shades in different colors (representing the two performed experiments for bar and line graphs, respectively). This will possibly also show that the altered peak at the last TMD is an over-interpretation. For the bar plots it would be also good to show them with

logarithmic y-axis scale to make easily visible what is up and what is down-regulated – this also then makes the extent of regulation equally visible in both directions (i.e., up and down).

---Thank you for your constructive comments. For RNA-seq and Ribo-Seq cpRPKM, we combined three biological replicates for *fpb1* and *pam68* and show it with standard deviations. The *lpa1* data was obtained for two biological replicates and we show it with average with two dots in the figure. Please refer to the new Fig. 6a and 6b and both figures are shown using a logarithmic y-axis scale.

For the distribution of ribosome footprints along the *psbB* ORF in the three replicates of WT, *fpb1*, and *pam68* mutant plants, we found that it is difficult to combine them together. The relative height of the peaks of various genotypes are comparable within one experiment (eg. the first (Fig. 6c) and second (Fig. 6d) replicates in the first experiment or the third (Fig. 6e) in the second experiment). However, the relative height of some peaks is variable between two experiments (eg. the peaks III and IV of *psbB* in Fig. 6c-e). These imply that Ribo-Seq experiment might be influenced by many factors, such as the light intensity during the material harvesting, quality control of the experimental process. In the revised manuscript, we show three replicates within one figure (Fig. 6c-6e) and the calculated values of the relative peak height are shown in Supplementary Fig. S9. Please refer to the new Fig. 6 and L641-L657.

8. Similarly, you state that “after normalization of ribosome footprint abundance, the fifth peak in *fpb1* and *pam68* is significantly higher than in WT in all three replicate samples (Figs. 6c and 6d, Supplementary Fig. 7b).” How was this significance statistically tested and is it the only found significant change in ribosome pausing between the different mutants and the WT throughout the whole chloroplast genome? A genome-wide analysis is needed here with statistical tests. I am afraid that you otherwise over-interpret the data at this point (although I have to admit that I am very much in favor of your hypothesis!).

---This is a really constructive comment! In the revised manuscript, we performed a genome-wide analysis of the change in ribosome pausing between the different mutants and WT. A threshold was set in this analysis, in which the ribosome footprints reads at each position along the ORF less than 500 were not included in the assay (a total of 38 genes). The relative heights of the major peaks of the remaining 50 genes were calculated and the results are summarized in Supplementary Fig. 9 and Supplementary Data 3.

The genome-wide analysis showed that, for the relative height of the *psbB* fifth peak (*psbB-V*), the ratios of *fpb1* and *pam68* to WT are 1.310 ± 0.086 and 1.420 ± 0.041 , respectively. Only two peaks (*psbA-V* and *psbZ-I*) exhibiting higher relative heights in the *fpb1* and *pam68* mutants than the fifth peak of *psbB* (Supplementary Fig. 9). For the fifth peak of *psbA* (*psbA-V*), a higher relative height was observed in *fpb1*, *pam68*, *lpa1* as well as in our previously reported *rbd1* mutant (Supplemental Dataset 5; Che et al., 2022). This peak is localized in the region of *psbA* mRNA encoding the third TMD, suggesting that the ribosomes pause during *psbA*

translation when the third TMD of D1 emerges from the ribosomal tunnel. Interestingly, the synthesis rate of D1 is reduced in all these four mutants (Fig. 2b; Peng et al., 2006; Che et al., 2022). It is possible that *psbA* translation at this position is secondarily feedback-regulated in the absence of FPB1 and PAM68. Alternatively, FPB1 and PAM68 are also directly involved in the D1 synthesis, a possibility we cannot exclude. For the first peak of *psbZ* (*psbZ*-I), only *fpb1* showed a relative higher height compared with WT.

In addition, the genome-wide analysis also detected several peaks with relative slightly higher heights (*psbD*-VI, *psaB*-I, *petA*-VI, *rps11*-I, and so on) or lower heights (such as *psbA*-III, *psbD*-1, *petB*-I, *atpH*-II, and so on) in the *fpb1* and *pam68* mutants compared with WT. This is possibly due to the limitations of the Ribo-Seq technology but we cannot exclude the possibility that translation of these genes is directly or indirectly affected in these mutants. However, our genetic results and biochemical results indicate that *psbB* is the main target of FPB1 and PAM68. Thus, according to the genome-wide analysis, we have toned down some of the statements in the revised manuscript. Thanks to the reviewers' comments we believe that the revised version of this part has been significantly improved. Please refer to L697-716.

9. Please also compare your WT *psbB* pausing sites to those in published datasets of Arabidopsis and other species for which Ribo-seq data are available (e.g., maize). If the peaks (i.e., pausing sites) are related to features of the nascent peptide (e.g., TMDs as you speculate) they should be conserved because the CP47 protein is highly conserved in the green lineage. Since the mRNA sequence is less conserved, mRNA features that lead to pausing may be less conserved in different species.

---Good suggestions. We compared our results with two Ribo-seq data from Arabidopsis and three Ribo-seq data from maize published by Chotewutmontri and Barkan (2018) and the results are shown in Supplementary Fig. 10. We found that the *psbB* peaks I, II and V are well detected in Arabidopsis and maize in Chotewutmontri and Barkan (2018), suggesting that *psbB* pausing at these sites is conserved in these two species. Probably due to the different protocol used in Ribo-seq analysis, the *psbB* peaks III and IV are not well detected in Arabidopsis or in maize in Chotewutmontri and Barkan (2018).

The alignment of the CP47 and *psbB* sequences shows that not only the CP47 protein but also the *psbB* sequences from Arabidopsis and maize are well conserved (Supplementary Fig. 11). Thus, conserved *psbB* mRNA features in different species may also contribute to the ribosome pausing at specific sites during translation. We briefly discuss this possibility in the revised manuscript. Please refer to L725-769.

10. Figure 7: In their model step 1 ff. it would be good if there would be at least two ribosomes on the *psbB* mRNA. The downstream ribosome (with its nascent CP47) would localize the mRNA to the position where its translation is needed (close to the thylakoid membrane at PSII biogenesis sites), whereas the upstream ribosome is the one you show already (but the upstream ribosome needs to move from step 1 to step

2 to produce the nascent peptide that is bound by the targeting machinery). The way you show it now would represent the very first translation initiation on a *psbB* mRNA after or during transcription, which is not the most likely case to observe in chloroplasts – due to the long half-life of transcripts (it is more likely to see translation repeated translation initiation on a transcript that is already translated by further downstream ribosomes). Between step 4 and 5 make the arrow in a way that it connects the steps.

---Thank you for the comments. We agree with this reviewer that the translation initiation is very complicated as other steps. The important point of our work is to show the different stages of the translation of the *psbB* mRNA with the sequential insertion of the nascent chain of CP47 into the thylakoid membrane. Since our work does not focus on the very first translation initiation step of CP47, we would like to ignore this step in our model. In this case, our model starts from the nascent peptide that is bound by the targeting machinery (Step II in the old version of the manuscript). To make this model as clear and simple as possible so that it can easily be understood by the reader we prefer to show only one ribosome. We introduced a new model in the revised manuscript and slightly modified the description in the new figure legend. Please refer to L860-870.

11. I did not find the information where the sequencing data is deposited.

---The information is provided in the revised manuscript. Please refer to L1124-L1126.

Minor points:

1. Line 122 ff.” a total of 37 chloroplast-encoded proteins are intrinsic thylakoid proteins and 19 of them have been proposed to be co-translationally inserted into the membrane (Zoschke and Barkan, 2015)“ – From my understanding the data in the cited paper demonstrate the co-translational insertion (not only “proposes” it).

---Thanks. We modified our expression as “a total of 37 chloroplast-encoded proteins are intrinsic thylakoid proteins and 19 of them have been demonstrated to be co-translationally inserted into the membrane”. Please refer to L96.

2. Suppl. Figure 4: 1. It would be nice to label the size of the bands or, even better, provide ladder sizes so that readers (and reviewers) can decide which transcript isoform/s is/are shown. 2. It should be *psbE/F/L/J*.

---We labeled the RNA ladder size and corrected to *psbEFLJ* throughout the manuscript. Please refer to the new Supplementary Fig. 4a and L244-245.

3. line 441: circled instead of cycled.

---We corrected it throughout the manuscript. Please refer to L381.

4. line 674/75: “The large error bar with *psbM* mRNA is probably due to its small size resulting in a low number of RNA-Seq reads (Fig. 6a; Supplementary Table 2).” There are several other chloroplast-encoded small PSII subunits that do not show this

high standard deviation... Could there be a biological reason for this behavior (e.g., related to the PsbM localization at the interphase of the PSII dimer)? You may consider to set a threshold that excludes genes with low read number to not distract the focus from your major points with high standard deviation of psbM.

---Good comments. The read number for psbM is less than 200 in the samples of the first experiment, which is far less than the reads of other PSII genes. We now provide this information as: “For RNA-seq, the *psbM* gene was excluded because its number of reads is the lowest among PSII genes and less than 200 in the samples of first experiment”. Please refer to L645-647.

5. line 694 ff.:” The ribosome footprints showed a peak at 32-33 nucleotides (Supplementary Fig. 6a), which is consistent with the cytosolic ribosome footprints in Maize and the 31-nucleotide modal size reported previously...” Usually for Arabidopsis smaller footprint sizes are reported – so your samples are probably slightly “under-digested”. I do not think that this influences the interpretation of the data at all but it makes the previous statement wrong.

---Thank you for your comments. We deleted the information from other reports and just described our results in the revised manuscript. Please refer to L622-L623.

6. line 697 ff.: “The frequency of footprints with a 5’ end at the adenine nucleotide was about 40% (Supplementary Fig. 6b), which is lower than the cytosol ribosome footprints reported in Maize (~50% in Chotewutmontri and Barkan, 2016).” I do not get the point – maybe explain a little bit better how you infer from the nature of the 5’ end nucleotide to the triperiodicity. Also maize (starting with lower case letter).

---Similar to the last response, we deleted the information from other reports and just described our results. We also briefly explained the nature of 3-nucleotide periodicity as: “The three-nucleotide periodicity corresponds to the translocation of ribosomes along the mRNA three nucleotides at a time³³. Our data show the 3-nucleotide periodicity expected for ribosome prints and the footprints coverage is largely restricted to the open reading frames (Supplementary Fig. 7b, c).” Please refer to L623-L627.

7. Supplementary Figure 6: if the number of replicates is below 3, no standard deviations should be shown. Subfigure c: state which set of gene was used for this meta-analysis – only cp genes or only nuclear genes or all genes, genes above a specific expression threshold, how many genes?

---This figure is Supplementary Fig. 7 in the revised manuscript. We removed standard deviations and only the means of two replicates are shown. In the subfigure c, all genes (22895 detected genes) with unique mapped reads of ribosome footprints were analysed. Please refer to L680-688.

8. line 744: you refer to (Kim et al., 1991) which analyzes ribosome pausing in psbA but not psbB.

---Thank you for your comments. We deleted this citation and slightly modified the

text in the revised manuscript as “Five major peaks were detected in the distribution (Figs. 6c, d, e), suggesting the positions of paused ribosomes along the mRNA.” Please refer to L660-L662.

9. line 811 ff.: PsbH was also suggested to be involved in CES in Chlamy by Trösch et al. 2018.

---Yes, we cited this paper in the revised manuscript as “In addition, PsbH was also suggested to be involved in CES (for control by epistasy of synthesis) regulation of *psbB* in Chlamydomonas (Trösch et al., 2018).”. Please refer to L795-797.

10. Methods line 982 ff.: please make clear which material (including detailed growth conditions and developmental stage etc. pp.) was used for which experiment!

---Thanks for your reminder. We described the information of material in both the Methods and the figure legends in detail. Please refer to the red character in Methods and the figure legends.

REVIEWERS' COMMENTS

Reviewer #1 (Remarks to the Author):

The manuscript of Lin Zhang et al. has been carefully revisited and many questions properly responded. I would like to see this work published after addressing a few remaining points. Data are sufficient to conclude that both FPB1 and PAM68 are needed for the synthesis of CP47 and both these factors act in the vicinity of translocon machinery. To demonstrate these interactions with an acceptable confidence level is indeed challenging. The content of nascent CP47 in complex with FPB1 and PAM68 must be extremely low and difficult to detect. Full MS dataset indicate that co-IP with anti-FPB1 and anti-PAM68 are contaminated by fully-assembled PSII (CP43 or PsbP are present in relatively high content) rather than to contain unassembled CP47 and this is not surprising. ALB3 (and VIPP) appear to me as the only good hits of Co-IP and it would not be shame to frankly say that. Authors made however a good job and Co-IP assays after crosslinking look convincing now, particularly in combination with Y2H.

I struggle a bit to accept the presented model (Fig. 7) but it might just problem in terminology used. Authors propose the role of the FPB1/PAM68 couple in insertion (integration) of CP47 helices into the membrane. However, transmembrane helices are already inserted into membrane at the moment they are laterally released from the Sec translocase; the ALB3 associated with SecY serves rather a foldase role, not insertase. What role the FPB1/PAM68 would play in the process of insertion into membrane? It is likely that cpSecA is involved in the translocation of the large E loop of CP47 but there is no clear evidence that the FPB1/PAM68 interact directly with cpSecA. And it is indeed a question why such interactions would be needed to stabilize(?) or activate(?) cpSecA. For the sake of accuracy, the model that the SecA (ATP hydrolysis) provides a driving force for translocation is obsolete (see e.g. EMBO J, 2024, 43:1-13) and SecA and ribosome cannot associate, at the same moment, with a single SecYE as shown in Fig. 7.

As an alternative model, both FPB1/PAM68 could facilitate folding of CP47, perhaps in collaboration with ALB3. Misfolding of CP47 loops and helices can affect the translation pausing, but as translational pausing is often very robust (PNAS, 2016 113:E829), the observed effect is rather subtle. Misfolded CP47 is likely recognized for degradation by Sec-associated proteases before the protein translation is finished, which can further disturb the activity of ribosome. This model is also consistent with results in cyanobacteria because chlorophyll molecules must play crucial role in CP47 folding. Chlorophyll cofactors are so structurally important for CP47 that is highly unlikely that their insertion and CP47 folding can be somehow decoupled (line 903-904). Results of Zoschke R, (Front Plant Sci, 2017 8:385), showing that the deficiency of chlorophyll does not alter the pausing, are in fact in line with the possibility that the misfolding/degradation of CP47 has only a limited effect on the activity of psbB mRNA-associated ribosomes. I don't push this model but it could be at least considered.

Recently, FPB1 (named DEAP2) has been connected with PAM68 by another group (Plant Physiol, 2023, 193:1970). This work needs to be at least referred. Intriguingly, authors claim that PAM68 has no effect on the synthesis of CP47 though their own data (Fig. 6) show weak synthesis of CP47 in the pam68 mutant line.

Minor comments:

Line 542 "Its component SecY1 and Alb3 were readily detected in the FPB1- and PAM68-precipitates but not in the Pre-serum and YCF4-precipitate by MS analysis (Supplementary Data 1)." I cannot find SecY in dataset 1, only very small amount of SecA and FtsY.

Line 799: I think that CtpA must act before the attachment of PsbP and PsbO. Why the formation of RC47 accelerates the pD1 maturation is not clear but it could be due to a partial displacing of HCF136 after the CP47 binds (see Nat Commun, 2023, 14:4681)

Lane 820 SLL0933, should be SII0933

Reviewer #3 (Remarks to the Author):

In the revised manuscript, the authors nicely addressed all questions I raised in my previous review! Since the very same factor was very recently described by another group, it is necessary to comparatively discuss your results with their results <https://pubmed.ncbi.nlm.nih.gov/37555435/>. Also, you may think about the name of the factor: is there a good reason to have a distinct name for it (according to your data) or should you rather name it also DEAP2 (the name given by Armbruster and coworkers).

I am looking very much forward to see your results published in Nature Communications!

Dear Editor,

We wish to thank the reviewers for their constructive comments and criticism. Let me answer the different points raised by the reviewers.

Reviewer 1

R1. The manuscript of Lin Zhang et al. has been carefully revisited and many questions properly responded. I would like to see this work published after addressing a few remaining points. Data are sufficient to conclude that both FPB1 and PAM68 are needed for the synthesis of CP47 and both these factors act in the vicinity of translocon machinery. To demonstrate these interactions with an acceptable confidence level is indeed challenging. The content of nascent CP47 in complex with FPB1 and PAM68 must be extremely low and difficult to detect. Full MS dataset indicate that co-IP with anti-FPB1 and anti-PAM68 are contaminated by fully-assembled PSII (CP43 or PsbP are present in relatively high content) rather than to contain unassembled CP47 and this is not surprising. ALB3 (and VIPP) appear to me as the only good hits of Co-IP and it would not be shame to frankly say that. Authors made however a good job and Co-IP assays after crosslinking look convincing now, particularly in combination with Y2H.

A1. We appreciate these thoughtful comments. The content of nascent CP47 in complex with FPB1 and PAM68 is indeed low and difficult to detect. We agree with the comments that the Co-IP samples were contaminated by trace amount of fully-assembled PSII and we mention this in the revised manuscript (L381-384). As this reviewer wrote, this kind of contamination is inevitable and does not affect the detection of the main targets of FPB1 and PAM68, such as Alb3. We feel that the Co-IP with the crosslinking data of Fig. 5b and Y2H of Fig. 5d show convincingly specific interactions between CP47 and FPB1 and PAM68.

R2. I struggle a bit to accept the presented model (Fig. 7) but it might just problem in terminology used. Authors propose the role of the FPB1/PAM68 couple in insertion (integration) of CP47 helices into the membrane. However, transmembrane helices are already inserted into membrane at the moment they are laterally released from the Sec translocase; the ALB3 associated with SecY serves rather a foldase role, not insertase. What role the FPB1/PAM68 would play in the process of insertion into membrane? It is likely that cpSecA is involved in the translocation of the large E loop of CP47 but there is no clear evidence that the FPB1/PAM68 interact directly with cpSecA. And it is indeed a question why such interactions would be needed to stabilize(?) or activate(?) cpSecA. For the sake of accuracy, the model that the SecA (ATP hydrolysis) provides a driving force for translocation is obsolete (see e.g. EMBO J, 2024, 43:1-13) and SecA and ribosome cannot associate, at the same moment, with a single SecYE as shown in Fig. 7.

A2. We agree with these comments and the question is left open whether FPB1 and PAM68 are also involved in the folding of the proteins. Concerning SecA we agree that rather than a direct coupling of SecA ATP binding/hydrolysis to SecYEG channel motions, the SecA ATPase cycle regulates SecYEG opening and closing as shown by Crossley et al (2024). Fig. 7 has been redrawn to take into account the concerns of the reviewer regarding SecA. We also mention this in the revised manuscript (L625-626).

R3. As an alternative model, both FPB1/PAM68 could facilitate folding of CP47, perhaps in collaboration with ALB3. Misfolding of CP47 loops and helices can affect the translation pausing, but as translational pausing is often very robust (PNAS, 2016 113:E829), the observed effect is rather subtle. Misfolded CP47 is likely recognized for degradation by Sec-associated proteases before the

protein translation is finished, which can further disturb the activity of ribosome. This model is also consistent with results in cyanobacteria because chlorophyll molecules must play crucial role in CP47 folding. Chlorophyll cofactors are so structurally important for CP47 that is highly unlikely that their insertion and CP47 folding can be somehow decoupled (line 903-904). Results of Zoschke R, (*Front Plant Sci*, 2017 8:385), showing that the deficiency of chlorophyll does not alter the pausing, are in fact in line with the possibility that the misfolding/degradation of CP47 has only a limited effect on the activity of *psbB* mRNA-associated ribosomes. I don't push this model but it could be at least considered.

A3. We thank the reviewer for this additional possibility which we have incorporated in the Discussion. Please refer to L640-644 and L649-651.

R4. Recently, FPB1 (named DEAP2) has been connected with PAM68 by another group (*Plant Physiol*, 2023, 193:1970). This work needs to be at least referred. Intriguingly, authors claim that PAM68 has no effect on the synthesis of CP47 though they own data (Fig. 6) show weak synthesis of CP47 in the *pam68* mutant line.

A4. We have cited this work and comparatively discussed the results in the Discussion. Please refer to L518-525.

R5. Minor comments:

Line 542 "Its component *SecY1* and *Alb3* were readily detected in the FPB1- and PAM68-precipitates but not in the Pre-serum and YCF4-precipitate by MS analysis (Supplementary Data 1)." I cannot find *SecY* in dataset 1, only very small amount of *SecA* and *FtsY*.

A5. The *SecA1* protein can be found in L46 (Sheet 1), L813 (FPB1-Co-IP), and L458 (PAM68-Co-IP).

R6. Line 799: I think that *CtpA* must act before the attachment of *PsbP* and *PsbO*. Why the formation of RC47 accelerates the *pD1* maturation is not clear but it could be due to a partial displacing of HCF136 after the CP47 binds (see *Nat Commun*, 2023, 14:4681)

A6. We agree with these comments and modified the text. Please refer to L555-558.

R7. Lane 820 SLL0933, should be SII0933

A7. Corrected.

Reviewer 3

R. In the revised manuscript, the authors nicely addressed all questions I raised in my previous review! Since the very same factor was very recently described by another group, it is necessary to comparatively discuss your results with their results <https://pubmed.ncbi.nlm.nih.gov/37555435/>. Also, you may think about the name of the factor: is there a good reason to have a distinct name for it (according to your data) or should you rather name it also DEAP2 (the name given by Armbruster and coworkers).

A. We have cited this work and comparatively discussed the results in the Discussion (L518-525). Concerning the name for this PSII factor we feel that FPB1 is more appropriate. However, we wish to

redefine the FPB1 as Facilitator of PsbB biogenesis1, which directly points to the function of this protein.